# Investigating similarities and differences of the penultimate and last glacial terminations with a coupled ice sheet – climate model

Aurélien Quiquet[1] and Didier M. Roche[1,2]

[1]Laboratoire des Sciences du Climat et de l'Environnement, LSCE/IPSL, CEA-CNRS-UVSQ, Université Paris-Saclay, F-91191 Gif-sur-Yvette, France
[2]Earth and Climate Cluster, Faculty of Earth and Life Sciences, Vrije Universiteit Amsterdam, Amsterdam, the Netherlands

**Correspondence:** A. Quiquet (aurelien.quiquet@lsce.ipsl.fr)

**Abstract.**

Glacial terminations are marked by a re-organisation of the different components of the climate system. In particular, rapid ice sheet disintegration leads to multiple complex feedback loops that are still poorly understood. To further investigate this aspect, we use here a fully coupled Northern Hemisphere ice sheet – climate model to perform numerical experiments of the last two glacial terminations. We show that even if the first-order climate trajectory is similar for the two terminations, the difference in terms of solar insolation leads to important changes for the ice sheet – climate system. Warmer temperatures during the penultimate termination are compatible with higher sea level during the last interglacial period with respect to the Holocene. We simulate a last interglacial Greenland contribution to sea level rise of about 2 m of sea level equivalent. We also simulate warmer subsurface Southern Ocean, compatible with an additional contribution from the Antarctic ice sheet. In addition, even without considering freshwater flux to the ocean resulting from ice sheet melting, the two terminations display different Atlantic overturning circulation sensitivity, this circulation being more prone to collapses during the penultimate termination. Finally, with additional sensitivity experiments we show that, for the two terminations, the Northern Hemisphere insolation is the main driver for the ice sheet retreat even if vegetation changes have also to be taken into account to simulate the full deglaciation. Conversely, even though it impacts the temperature, greenhouse gas concentration change alone does not explain the amplitude of ice sheet retreat and only modulates its timing.

## 1 Introduction

The geological record of the Quaternary is characterised by climatic oscillations alternating from cold - low sea level glacial periods to warm - high sea level interglacial periods. Over the last million years these oscillations display a remarkably large amplitude and are strongly asymmetric (Lang and Wolff, 2011): the long (~80 ka) glacial periods show a general cooling trend before abruptly switching to a short (~10 ka) interglacial period. Thus, during glacial terminations, the global mean temperature can increase by 3 to 5 degrees (Annan et al., 2022) and the eustatic sea level rises by approximately 100 metres in about 10 ka (Lambeck et al., 2014; Spratt and Lisiecki, 2016). The study of glacial terminations can provide insights into the future deglaciation since it offers an unique opportunity to understand large scale ice sheet retreats under a warming climate and the

impact of these retreats on the global climate system.

Among the different terminations, the last glacial termination, here after referred as Termination I (TI), is the best documented. The ice sheets present their maximum volume between 26 and 20 kaBP (Lambeck et al., 2014; Gowan et al., 2021). From the last glacial maximum (LGM, hereafter 21 kaBP), the sea level gradually rises and approaches its modern value already around the middle Holocene (6 kaBP). Several abrupt events have marked this deglacial sea level rise. Notably, palaeo-coral

reconstructions show that circa 14.6 kaBP, during the so-called melt water pulse 1A (MWP-1A, Deschamps et al., 2012), the rate of sea level rise could have reached more than 5 metres per century. Such event suggests a large scale ice sheet collapse. Ice rafted debris concentration in marine sediments also offer an indirect indications of ice sheet changes. Notably, high concentration of such debris during the Heinrich event 1 (H1), circa 17 kaBP, suggests a massive release of icebergs in the North Atlantic (Heinrich, 1988; Hemming, 2004). In parallel to the ice sheet changes, the atmosphere also experiences large

and sometimes abrupt changes during the last termination. For example, Greenland temperature deduced from ice core records shows an abrupt warming event of about 10°C in a few decades at the onset of the Bølling-Allerød warm period (Buizert et al., 2014), synchronous with the MWP-1A. These interglacial conditions do not last long and are followed by a succession of cooling events, the most prominent one being the Younger Dryas at ~12.8 kaBP (Alley, 2000). On the other hemisphere, Antarctic ice cores display a gradual warming during the last termination, stalled during the so-called Antarctic cold reversal

(ACR) where the local air temperature displays a cooling trend. The onset of this period is also synchronous with the MWP-1A. In the ocean, marine sediments record fluctuations in the strength of the Atlantic meridional overturning circulation (AMOC), warmth conveyor to the North high latitudes (Böhm et al., 2015; Lynch-Stieglitz et al., 2014; Ng et al., 2018). At the LGM, the AMOC was probably weaker and shallower than today. It was eventually completely shutdown in the first phase of the glacial termination, at the time of H1, for about 3.5 ka (Böhm et al., 2015). The AMOC switched back to an active state during the

warm Bølling-Allerød but slowdown again during the cold Younger Dryas, without being completely shutdown. From the end of the Younger Dryas the AMOC gradually increased to its modern state.

The penultimate glacial termination, hereafter referred as Termination II (TII) is also relatively well documented, even tough there are less data and they are associated with larger dating uncertainties. The penultimate glacial maximum (PGM, hereafter

140 kaBP) could have presented a similar eustatic sea level to the one of the LGM (Rabineau et al., 2006; Rohling et al., 2017). However the geometry of the different ice sheets was probably drastically different. In particular, the Eurasian ice sheet could have been more extended to the East during the PGM (Svendsen et al., 2004; Lambeck et al., 2006; Colleoni et al., 2016; Batchelor et al., 2019; Pollard et al., 2023) suggesting a probable smaller North American ice sheet. Nevertheless, the maximal expansion of the Eurasian ice sheet might have occurred significantly earlier than the PGM (Hughes and Gibbard, 2018; Pol-

lard et al., 2023) and precise reconstruction of the PGM ice sheets is still lacking. From the PGM, the ice sheets retreated until 121 kaBP to produce a global sea level that might have culminated at 6 to 8 m above its present-day value (Dutton et al., 2015), even though recent estimates suggest a smaller peak sea level highstand, ranging from 1 to 5 m (Dyer et al., 2021). The ice sheet evolution through TII is less constrained than during TI since the proxy for palaeo extents have been generally scrapped

away during the last glacial period where it lays inbound the LGM extent. However, similarly to the last termination, the ice sheet retreat was punctuated by abrupt accelerations, similar to the MWP-1A (Stoll et al., 2022). Notably, a massive Heinrich event, H11, occured at about 132 kaBP so relatively late in the glacial termination (Obrochta et al., 2014) with respect to H1. Also, as for TI, proxy for the AMOC suggest a shutdown, or largely reduced, oceanic circulation (Böhm et al., 2015) during the penultimate glacial termination. However, the AMOC might have remained in a shutdown state for about 7 ka during TII, twice as long as for TI (Böhm et al., 2015; Deaney et al., 2017). To date, perturbed basal ice at the bottom of Greenland ice cores does not allow for a continuous reconstruction of atmospheric temperature evolution before 123 kaBP (NEEM community members, 2013). The Antarctic temperature evolution through TII does not present an equivalent of the ACR as it shows a gradual increase, culminating at 128 kaBP. Other types of records, such as speleothems or oceanic sediment data, display abrupt changes, concomitant with oceanic changes (Martrat et al., 2014; Govin et al., 2015; Cheng et al., 2016).

In summary, the last two glacial terminations display significant differences. In terms of ice sheet disintegration, there are some proxy data evidence for a higher rate of mass loss during TII with respect to TI (Carlson, 2008; Stoll et al., 2022; Grant et al., 2014). This higher loss rate might explain the long (~7 ka) period of weak AMOC across TII (Böhm et al., 2015; Deaney et al., 2017). A feature that significantly differs from the several shorter events during TI (McManus et al., 2004). If speleothem and oceanic records suggest that H11 share similar large scale characteristics with H1 or the Younger Dryas, these events largely differ in terms of timing of their occurrence during the termination (Martrat et al., 2014; Govin et al., 2015). In terms of ice sheet geometries, apart from the fact that they were different for the two glacial maximums (Svendsen et al., 2004; Pollard et al., 2023), the geometry changes through the terminations cannot be easily compared due to the lack of strong constraints for TII.

If the changes in term of ice sheets, atmosphere and ocean are getting better documented, the causal chain of events during terminations has yet to be formalised. To this aim, numerical models are powerful tools to explore hypotheses, to quantify the respective importance of feedbacks or to study the similarities and differences between different periods. There is now a relatively extensive literature about numerical experiments of the last termination. However, most of the time this literature consists of simulations where the ice sheet changes are not interactively coupled but prescribed (e.g. Menviel et al., 2011; He et al., 2013; Gregoire et al., 2016; Obase and Abe-Ouchi, 2019; Kapsch et al., 2022). This has the advantage to use an ice sheet reconstruction that is constrained by the palaeo data but it prevents the study of ice sheet climate feedbacks. An alternative has been to use an asynchronous coupling in which the ice sheet changes are computed offline for a given timespan and feed back later in the climate model (Abe-Ouchi et al., 2013; Heinemann et al., 2014). This strategy allows for numerically cheaper simulations since the climate model does not have to run transiently for the whole simulated period. To date, synchronously coupled simulations of the glacial termination have been performed only with the CLIMBER-2 and iLOVECLIM climate model of intermediate complexity (Charbit et al., 2005; Willeit and Ganopolski, 2018; Quiquet et al., 2021). All these simulations have contributed to a better understanding of the last termination even though a few open questions remain: i- the millenial scale abrupt variability is generally underestimated or is linked with abrupt changes in AMOC; ii- the different models show very

different AMOC states in the past (Kageyama et al., 2021) and; iii- the sensitivity of the AMOC to freshwater flux is generally too strong. There are less numerical simulations of TII. Recent GCM experiments have shown that the late and prolonged Heinrich event H11 lead to a major difference between TI and TII since it induced a prolonged AMOC shutdown state during TII (Clark et al., 2020; Obase et al., 2021). This AMOC shutdown late in the glacial termination could have facilitated the Antarctic ice sheet retreat since it would have been associated with sub-surface warming in the southern high latitudes (Clark et al., 2020).

In this paper we aim at exploring the similarities and differences of TII with respect to TI. We use a fully coupled Northern Hemisphere ice sheet – climate model to quantify the interconnected evolutions of ice sheets, atmosphere and ocean. Using a relatively simplified setup, we do not aim to precisely match the available proxy data but instead we aim at better understanding the role of external forcings (orbital configuration and greenhouse gas concentration) on glacial terminations. Sec. 2 describes our model and the different numerical experiments performed. In Sec. 3 we first present the simulated climate during the glacial maxima, LGM and PGM, before discussing the climate and ice sheet evolutions through TI and TII. This section also presents the simulated last interglacial climate and sea level rise and investigates the respective role of external forcings and internal feedbacks for the two terminations. We discuss our modelling assumptions with respect to the literature in Sec. 4. Finally, our findings are summarised in Sec. 5.

## 2   Methods

### 2.1   Models

We use the iLOVECLIM Earth system model of intermediate complexity version 1.1.5. iLOVECLIM is a fork from the LOVE-CLIM model (Goosse et al., 2010) with which it shares the main components, i.e. ocean, atmosphere and vegetation. The oceanic model, CLIO, is a general circulation - sea ice model that uses a 3° resolution and 21 vertical layers. The atmospheric model, ECBilt, is a quasi-geostrophic atmospheric model that runs on a T21 spectral grid (approximatively 5.6° resolution). The model includes additional ageostrophic terms to improve the atmospheric circulation in the Tropical region (Opsteegh et al., 1998). The vegetation model, VECODE, is a reduced-form dynamic global vegetation model that represents two plant functional types (trees and grass). The model has been used previously for a wide range of climatic applications. It has notably been shown capable of reproducing the changes on orbital timescale of the Asian monsoon dynamics Caley et al. (2014). It has also been used to study the mechanisms at play for the glacial deep ocean circulation (Lhardy et al., 2021) or during the last deglaciation (e.g. Renssen et al., 2015; Quiquet et al., 2021; Bouttes et al., 2023). For this work, we use the optional ice sheet model component GRISLI (Quiquet et al., 2018a) which is fully coupled to the rest rest of the climate model (Roche et al., 2014; Quiquet et al., 2021). GRISLI is a hybrid 3D thermo-mechanically coupled ice sheet model that solves the shallow approximations of the Stokes flow equations. The ice sheet model resolution is 40 km×40 km.

The bi-directional coupling of the ice sheet model to the atmospheric model ECBilt is performed through an interactive online downscaling at the ice sheet model resolution (Quiquet et al., 2018b). This downscaling consists on computing temperature and precipitation on the fine-scale GRISLI orography at each time step of the atmospheric model (4 hours). Surface mass balance of the ice sheet is defined as the difference between accumulation (solid precipitation) and ablation (melt, $M_s$), computed with an insolation - temperature - melt model (ITM, van den Berg et al., 2008):

$$M_s = max\left(\frac{dt}{\rho_w L_m}\left(\left(1-\alpha\right)SW_s + c_{rad} + \lambda T_s\right), 0\right) \tag{1}$$

With $T_s$ is the near surface air temperature, $SW_s$ is the shortwave radiation at the surface, $\alpha$ is the surface albedo, $\rho_w$ is the density of liquid water and $L_m$ is the specific latent heat of fusion. $\lambda$ and $c_{rad}$ are empirical parameters. While the $\lambda$ parameter is generally set to 10 W m$^{-2}$K$^{-1}$, $c_{rad}$ displays a wide range of values in the literature, ranging from about -120 to -40 W m$^{-2}$ (Pollard, 1980; van den Berg et al., 2008; Robinson et al., 2010). Given the fact that $c_{rad}$ is less constrained, we apply geographical corrections to this parameter to indirectly correct for the *iLOVECLIM* biases:

$$c'_{rad} = c_{rad} \times \left(1 + bias_f * T_{bias}\right) \tag{2}$$

With $T_{bias}$ the annual temperature bias with respect to ERA-interim (Dee et al., 2011). $bias_f$ is a correction factor that is set to 0.1 K$^{-1}$ so that a +10° K bias leads to $c_{rad}$ =-80 W m$^{-2}$ (instead of the reference value of -40 W m$^{-2}$). In practice, this geographical correction leads to a reduction of $c_{rad}$ in North America where there is a large warm bias and an increase of $c_{rad}$ in Northern Europe.

Surface mass balance and surface temperature are integrated over one year to provide the atmospheric forcings needed by GRISLI. In turns, orography and ice mask in ECBilt are updated every year following the simulated changes by GRISLI.

For the ocean, we use CLIO temperature and salinity at each vertical level to compute sub-shelf melting following Beckmann and Goosse (2003). We extrapolate this field over the entire ice sheet domain using a near-neighbour algorithm. Finally, the sub-shelf melt of the vertical level just below the ice shelf draft is applied to GRISLI. We also account for the freshwater flux to the ocean that results from ice sheet melting and iceberg calving. The freshwater flux due to surface melt is routed to the nearest ocean grid point using the routing scheme embedded in ECBilt. The calving flux is transferred to the nearest ocean grid point since the iceberg module is not activated here. These two fluxes are provided by GRISLI every year and are equally redistributed during the year in CLIO.

Here, model setup and parameter values are the same as in Quiquet et al. (2021). The main parameters are listed in Tab. S1.

## 2.2 Experimental setup

### 2.2.1 Boundary and initial conditions

The experiments discussed here for TI are the coupled ice sheet – climate model simulations covering the last 26 kaBP from Quiquet et al. (2021). For these simulations, the initial climate conditions and ice sheet geometries were obtained using uncou-

pled simulations. We first run the climate model for 3,000 years with prescribed ice sheet reconstructions (GLAC-1D, Tarasov et al., 2012; Tarasov and Peltier, 2002; Briggs et al., 2014) using fixed 21 kaBP orbital configuration (Berger, 1978) and greenhouse gas forcings (Lüthi et al., 2008). The last hundred years of this climate spin-up is used to derive climatological climate forcings required by the ice sheet model. We used these forcings to run stand-alone ice sheet model simulations for 200-kyr to reach equilibrium. The spun-up ice sheet and climate states were then used as initial conditions for our coupled simulations.

For TII, we follow the same methodology. We first run a glacial equilibrium of 2,000 years (starting from the LGM spin-up) with prescribed ice sheets using fixed 140 kaBP orbital configuration (Berger, 1978) and greenhouse gas forcings (Lüthi et al., 2008). This date corresponds to the minimum of Northern Hemisphere insolation and carbon dioxide concentration and will be considered in the following as representative of the PGM. The ice sheets for this PGM simulation are not interactive and they are fixed to their spun-up geometry at 21 kaBP of Quiquet et al. (2021). The climate obtained at the end of this PGM simulation is used as initial condition to all the subsequent TII transient experiments. The interactive ice sheets are activated for the transient experiments, starting from their 21 kaBP spun-up geometry. In doing so, the transient experiments of TI and TII differ by the forcings (insolation and greenhouse gas concentration) and the climatic initial state (LGM vs. PGM) but they share the same ice sheet initial state.

All transient TII experiments start at 142 kaBP. This choice is motivated by the fact that summer insolation in the Northern Hemisphere and carbon dioxide concentration are close to each other at 142 kaBP and at 26 kaBP, starting date of TI in Quiquet et al. (2021). In addition, from these dates onwards, the two insolation curves follow a similar evolution in time for the two terminations, both peaking 15 ka later (Fig. 1). However, we acknowledge that this is a somewhat arbitrary choice since, for example, the Southern Hemisphere insolation at 142 kaBP is substantially different to the one at 26 kaBP. Overall, there is nonetheless a well-preserved synchronicity in the forcings (North and South insolation as well as greenhouse gas concentration) over 142-116 kaBP and 26-0 kaBP. In addition to the orbital configuration and the greenhouse gas concentration, the eustatic sea level (Waelbroeck et al., 2002) is an other external forcing required by our model. It is used by the ice sheet model and can affect grounding line dynamics. The bathymetry, i.e. land mask and ocean depth, in the climate model remain fixed to the LGM bathymetry used in Quiquet et al. (2021). The impact of bathymetry on the climate trajectory has been extensively discussed in Bouttes et al. (2023).

All transient experiments span 26 ka, i.e. 26-0 kaBP for TI and 142-116 kaBP for TII.

### 2.2.2 Description of the experiments

Quiquet et al. (2021) identified that the freshwater flux resulting from ice sheet melting has large consequences on the Atlantic overturning circulation during the last glacial termination. iLOVECLIM was then assessed to be too sensitive to freshwater fluxes since with realistic fluxes, i.e. in the order of magnitude of the data-based eustatic estimates, we simulated a complete

and irreversible AMOC shutdown in the course of TI. For this reason, we consider in the following two reference experiments: with and without accounting for the freshwater release to the ocean due to ice sheet melting. For consistency with the TI experiments we also performed TII experiments with a division by two and by three of the amount of the freshwater fluxes.

In addition to these reference experiments, we also perform sensitivity experiments that use an acceleration factor of 5 for the forcings. In doing so we reduce the computing time by a factor 5 (5200 computed years instead of 26000) while covering the same temporal timespan. In these accelerated experiments, there is a decoupling factor of 5 for the coupling with the ice sheet model which is run 5 years every year of the rest of the climate model. In these accelerated experiments the freshwater flux resulting from ice sheet melting is discarded since we cannot preserve both the amplitude and the rate of the flux at the same time.

Accelerated experiments are first used to assess the sensitivity of the simulated TII to the initial ice sheet geometry. Our initial ice sheet geometry for our TII experiment is the same as for the TI experiment. This is a modelling simplification since it is unlikely that the configuration of the Northern Hemisphere ice sheets was identical for the two previous glacial maximums. To explore this model assumption, we elaborated alternative PGM ice sheet geometries. To generate these we run new stand-alone ice sheet model simulations using different SMB forcings to the ones used to generate the LGM ice sheet spin-up. The new SMB forcings were obtained by running the climate model for 100-yr simulations with regionally modified $c_{rad}$ parameter in the melt equation of the ITM. In the reference model, this parameter is locally adjusted to indirectly correct for the temperature bias in the model. To obtain alternative SMB we apply regional modifications to this temperature bias. More specifically, we reduce the bias correction in North America in order to generate higher surface melt rates since the temperature bias is positive in this region. In Eurasia, we impose a fixed artificial positive bias so that the $c_{rad}$ gets reduced to produce less melt. More information on these modifications is available in the supplement (Supp. Text 1). These artificial SMB modifications are only used to produce alternative ice sheets with GRISLI stand-alone simulations but they are removed for transient coupled simulations. The alternative ice sheet geometries consist in a reduced North American ice sheet by about 6% in volume with respect to the LGM (about -2.0 $10^6$ km$^3$) and a larger (+36% volume, about +2.1 $10^6$ km$^3$) or much larger (+71%, about +4.2 $10^6$ km$^3$) Eurasian ice sheet. The first alternative (larger Eurasian ice sheet) does not change the total ice volume stored on land while the second (much larger Eurasian ice sheet) corresponds to an increase of about 5 m of sea level equivalent of this volume. The alternative Eurasian ice sheets display a larger extent towards the East more in agreement with the palaeo data (Svendsen et al., 2004). These experiments serve to quantify the sensitivity of our simulated deglacial climate and ice sheet trajectories to the ice sheet glacial geometry.

We also use the accelerated experiments to quantify the respective role of the external forcings (greenhouse gas concentration and orbital configuration) and some internal feedbacks (ice sheets and vegetation). These sensitivity experiments use either fixed greenhouse gases (at 142 and 26 kaBP for TII and TI, respectively), fixed orbital configuration (also at 142 and 26 kaBP for TII and TI, respectively), fixed ice sheets (at their initial state) or fixed vegetation (at its initial state as well). While one aspect of the setup is fixed, the rest evolves as in the reference experiments. These series of experiments are used to isolate

the effect of the two major forcings of our setting (orbital configuration and greenhouse gas concentration) and the two major internal feedbacks (ice sheet and vegetation). These experiments are run both for TII and TI.

## 3   Results

### 3.1   Similarities and differences of the penultimate and last glacial maximums

The aim of this section is to analyse the spun-up glacial climates used as initial conditions of the transient experiments of TI and TII. Both spun-up climates have been obtained by running a 2 ka simulation with prescribed and fixed orbital configuration, greenhouse gas concentration and ice sheets. These forcings were selected at 140 kaBP and 21 kaBP, supposedly representative of the PGM and LGM, respectively.

The annual mean near-surface air temperature and precipitation simulated at the end of these equilibrium simulations are shown in Fig. 2. With respect to the simulated LGM, the PGM presents slightly cooler tropical region and slightly warmer southern high latitudes. In the Northern Hemisphere the pattern is more complicated, with a cooling in the vicinity of the Barents side of the Eurasian ice sheet and in East Siberia and a warming elsewhere. Nonetheless, these differences are small, generally lower than $\pm$ 1°C. The change in precipitation for the PGM with respect to the LGM are also relatively limited (less than 20 % change), except an important drying of West Africa and a wetting of Northern Australia. These changes in tropical precipitations are the results of the insolation seasonality differences between the LGM and the PGM (Supplementary Fig. S1). At the PGM, the decrease in boreal summer insolation reduces the West African monsoon, while an increase in austral summer insolation increases the monsoonal circulation over northern Australia. These changes in precipitation are amplified by the vegetation feedback, the model simulating a decrease in vegetation cover in West Africa but an increase in North Australia (Supplementary Fig. S2).

Over the ocean, the PGM glacial is generally warmer than the LGM, especially at high latitudes (Fig. 3). This warmer ocean at the PGM leads to a decreased sea ice thickness. This thinner sea ice at the PGM with respect to LGM can be largely explained by difference in the seasonality of insolation (Supplementary Fig. S1). For both hemispheres, there is an increase in insolation in the Autumn which tends to delay sea ice expansion and thickening. In terms of ocean dynamics, there is no significant change in the strength of the AMOC between the two glacial spun-up states (Supplementary Fig. S3). Only a slight weakening of deep water formation in the Austral ocean is simulated at the PGM related to a weaker seasonal sea ice amplitude (Supplementary Fig. S4).

## 3.2 Large scale climate change during the last two terminations

The evolution of selected integrated climatic variables through TII (142-116 kaBP) and TI (26-0 kaBP) terminations is shown in Fig. 4. At first sight, the two terminations look similar despite important differences. The major difference is that while the global mean temperature is very similar at the start of the termination experiments, it is rapidly becoming warmer during TII with respect to TI. For example, in the experiments including the freshwater flux resulting from ice sheet melting, the temperature at 137 kaBP when the eustatic sea level is still low (Fig. 1), is only reached at 16 kaBP, thus already well advanced in the last termination. This is directly the result of the forcing difference across the chosen time frame, with systematic larger values for the insolation (north and south) and greenhouse gas concentration during TII in the first part of the termination. The insolation curves display a larger amplitude during TII with respect to TI, with larger maximums but also lower minimums. This pattern explains why the peak temperature is higher and is reached sooner during TII but it is immediately followed by a gradual cooling, absent for TI.

The experiments that include the freshwater flux resulting from ice sheet melting display a halt in temperature increase in the course of the termination, from 133 to 131 kaBP for TII and from 13.5 to 11.5 kaBP for TI. This is related to the AMOC shutdown simulated within each termination. These shutdowns result in an important reduction of the heat transfer from low to high latitudes in the Northern Hemisphere. The prolonged AMOC shutdown state in the experiments that include freshwater flux is not in agreement with the palaeo record (Obrochta et al., 2014; Böhm et al., 2015). Interestingly, the TII experiment displays an abrupt AMOC recovery at 118 kaBP, i.e. during the progressive cooling corresponding to the end of the last interglacial period. It is very likely that the iLOVECLIM model has a marginally stable state under interglacial conditions (Jongma et al., 2007) and that a cold climate favours the active AMOC state. This is consistent with the oscillatory mode of the AMOC state already identified in iLOVECLIM under interglacial forcings (Friedrich et al., 2010; Kessler et al., 2020). The experiments that do not include the freshwater flux coming from ice sheet melting do not present an AMOC shutdown. In this case the two terminations look very similar in terms of AMOC evolution with a maximum in the course of the termination. For these two reference experiments (with and without freshwater flux), the main difference is that the changes during TII occur faster than during TI because of stronger external forcings. In Quiquet et al. (2021), we have shown that using a reduced freshwater flux across T1 we were able to produce abrupt variations in the AMOC while maintaining an active AMOC for the Holocene. This is no longer the case for T2 for which we systematically produce an AMOC collapse when freshwater flux are considered (even divided by a factor 3).

As for the AMOC, the evolution of sea ice extent in the two Hemispheres is drastically affected by the freshwater flux resulting from ice sheet melting. When this flux is discarded there is a progressive decrease of sea ice extent through both terminations. There is a quasi-synchronous sea ice minimum in both Hemispheres and for both terminations, reached at 128.5 kaBP for TII and 10 kaBP for TI. From this minimum, sea ice extent rises again but more rapidly towards the end of the last interglacial period than during the Holocene. When we take into account the freshwater flux feedback, changes in sea ice are more abrupt.

For TII, the AMOC decrease produces synchronous and opposite changes for the two Hemispheres: a rapid increase in the north and rapid decline in the south. This lasts for 1.5 ka before a progressive reduction in the Northern Hemisphere and rapid increase in the Southern Hemisphere. Since there is virtually no meridional heat transfer by the ocean at this time in this experiment, the difference in sea ice between the two Hemispheres is due to opposite trends in the atmospheric temperatures related to opposite insolation patterns. Overall, TII and TI sea ice temporal evolutions are very similar since they both respond firstly to freshwater flux and later to insolation changes. As for the AMOC shutdown, the TI sea ice evolution seems to lag the TII by approximatively 4 ka.

## 3.3 Last interglacial simulated climate

Marine sediment cores provide proxy based reconstructions of last interglacial sea surface temperature (SST) anomalies with respect to the pre-industrial that can serve to benchmark the model results (Capron et al., 2017). Data - model comparison for three snapshots through the last interglacial (130, 125 and 120 kaBP) is given in Fig. 5 for the experiment that account for the freshwater feedback to the ocean resulting from ice sheet melting. For this comparison, we use the summer SST reconstructions, computed in the model with respect to the simulated pre-industrial (0 kaBP). To account for the change in seasonality, we use the three warmest months for each Hemisphere, which translate into a shift of about 15 days for the 130 kaBP snapshot (7 days and none for 125 kaBP and 120 kaBP, respectively) in the Northern Hemisphere (none in the Southern Hemisphere). The model is in relatively good agreement with the available proxy data since it simulates a cooling of the North Atlantic at 130 kaBP and a warming at 125 kaBP. At 120 kaBP the model simulates a slight cooling of the North Atlantic when the proxy data suggest a more complex picture with both warming and cooling signals. There are more disagreements in the Southern Ocean. The model simulates a cooling during the austral summer for the three snapshots across the last interglacial period while the proxy data suggest a warming. The Southern Hemisphere SST in the model responds to the weaker Southern insolation for this three snapshots compared to its pre-industrial value. Several reasons could explain this disagreement. First, we do not consider Antarctic ice sheet changes in our setup. A West Antarctic collapse could affect the Southern Hemisphere climate since it will result into an important reduction of the ice mask and thus surface albedo. Second, the summer temperature proxy could reflect a sub-surface temperature rather than a sea-surface temperature. In fact, the model does simulate a small sub-surface warming in some places of the Southern Ocean that does not translate to a surface warming (Fig.6). Since we identified that the freshwater feedback resulting from ice sheet melting has a drastic impact on the ocean circulation, we also perform the data - model comparison for the experiment that does not account for this feedback. Such a comparison is shown in Supplementary Fig. S5. In this case, the Southern Ocean is warmer than when the freshwater flux is accounted for. However, the Southern Ocean SST anomalies remain generally negative, in contradiction with proxy reconstructions. Another major difference is that the 130 kaBP summer anomalies are much warmer when the freshwater flux is not accounted for. In this case, there is a strong disagreement with the proxy reconstruction. This suggests that, in our model, the North Atlantic cooling during the early interglacial is predominantly caused by an AMOC reduction triggered by freshwater flux. This result

is in agreement with previous findings from modelling experiments (Stone et al., 2016).

Compared to marine records, temperature change derived from ice cores has the advantage to display a continuous record with a high temporal resolution. If the Antarctic deep ice cores cover the entire last interglacial period, the longest continuous record in Greenland ice cores reaches back 123 kaBP. In Fig. 7 we show the simulated atmospheric temperature over Greenland and Antarctica across the two terminations, together with the proxy for temperature change. For the two terminations, the simulated temperature change over Greenland (more than 10 °C) is much larger than over Antarctica (about 2 °C). This difference among the two poles is consistent with proxy based temperature reconstuctions (Buizert et al., 2018, 2021) although with smaller temperature change in the model with respect to proxy-based reconstructions. A striking difference among the two terminations is that the last interglacial Greenland temperature remains above the Holocene temperature for about 8 ka. We simulate a temperature difference peaking 3 °C above the Holocene temperature circa 126 kaBP. This number, which includes the elevation change, is consistent with proxy-based estimates suggesting $5.2 \pm 2.3$ °C at North GRIP (Andersen et al., 2004; Landais et al., 2016). Another difference among the two terminations is the temperature overshoot during the last interglacial period which is absent for the Holocene. This overshoot occurs at 128 kaBP at EPICA Dome C, two thousand years before Greenland, a feature consistent with proxy reconstructions Landais et al. (2016).

## 3.4 Ice sheet evolution and the last interglacial highstand

The ice sheets of the Northern Hemisphere disappear sooner during TII with respect to TI (Fig. 8). While all the simulations start with the same ice sheets, the TI ice volume lags by approximately 3 ka the TII ice volume. This difference in timing is explained by the fact that the ice sheet volume is slightly increasing during the first 6 ka of TI, from 26 to 20 kaBP, while it decreases already at 138 kaBP, so 4 ka after the start of the TII experiment. However, the slopes of the deglacial ice volume curves are relatively similar, meaning that the retreat rates are not drastically different among the two terminations. It takes about 10 ka for both terminations for a complete desintegration of the North American and Eurasian ice sheets. In Fig. 9, we show the simulated ice sheets for selected snapshots of two terminations for equivalent dates after the start of the simulations (+5, +12, +14 and +26 ka). Already visible in Fig. 8, the ice sheets disintegrate faster during TII. However, for a given ice volume equivalent, the geometries of the ice sheets are very similar for the two terminations (Supplementary Fig. S6). This means that, in our model, changes in the forcings alone (orbital configuration and greenhouse gases) are not able to produce notable differences in the pattern of deglaciation when starting from identical ice sheets.

Even though our spatial resolution (40 km×40 km) is relatively coarse to have an accurate representation of the Greenland ice sheet, our simulations can be used to quantify its contribution to the last interglacial sea level rise. At the time of minimal ice volume during the last interglacial, circa 125 kaBP, the Greenland ice sheet is reduced in the West with respect to its simulated Holocene geometry. The ice volume difference corresponds to an equivalent of 1.9 m of sea level equivalent (m of SLE) when the freshwater flux due to ice sheet melting is accounted for. If this flux is discarded, the Greenland ice sheet contribution to sea

level rise during the last interglacial period is slightly larger, being 2.2 m of SLE, due to higher maximal Northern Hemisphere temperature in this experiment. These numbers are in general agreement with recent estimates (e.g. Dutton et al., 2015; Calov et al., 2015; Goelzer et al., 2016; Sommers et al., 2021).

We cannot quantify the Antarctic ice sheet contribution to sea level rise since it is not interactive in our experiments. Nonetheless we can compare the evolution of sub-surface oceanic temperatures for the two terminations (Fig. 11) since their difference most likely explains the Antarctic ice sheet contribution to the last interglacial. The major difference is that the austral sub-surface ocean is warmer during the penultimate glacial compared to the last glacial period. This is consistent with the SST difference shown in Fig. 3. The temporal change in Fig. 11 indicates that the sub-surface temperature is systematically higher during the whole duration of TII with respect to TI when the freshwater flux feedback on oceanic circulation is discarded. The freshwater flux leads to a more complex oceanic signal. The progressive decrease in the AMOC strength during TI leads first to a generalised sub-surface warming. But soon after its complete collapse the temperature starts to decrease. For TII the picture is slightly different. The AMOC early collapse starting around 134 kaBP produces a short-lived abrupt warming in the Weddel and Wilkes sectors at 133 kaBP while it produces a cooling for the Ross and Amundsen sectors. This difference between the two terminations is mostly explained by the difference in insolation in the Southern Hemisphere which tends to cool down the Southern Ocean during TII. Overall, the sub-surface ocean is generally warmer during TII and the temperature of the PGM is only achieved around 15 kaBP, thus well advanced in TI. A warmer sub-surface temperature during TII of about 0.1°C is simulated for the first part of the termination. This number is one order of magnitude below the projected sub-surface temperature change for the next century (Seroussi et al., 2020). This means that, in our model, the Antarctic retreat during the last interglacial could be the result of a prolonged small heat excess in the ocean rather than the result of an abrupt oceanic warming linked to AMOC changes. However this result might also be the consequence of our simplified setup in the Southern Hemisphere since, for both terminations, we do not account for Antarctic ice sheet changes (topography nor freshwater flux).

## 3.5 Accelerated sensitivity experiments: impact of initial ice sheet state and respective role of external forcings and internal feedbacks

In this section we present additional sensitivity analysis to complement our reference experiments presented earlier. These sensitivity experiments all use an acceleration factor in the forcings to save computational time and they thus differ from the reference non-accelerated experiments. Fig. 13 shows the simulated TII large-scale climatic indicators for the accelerated experiments using different initial ice sheet geometries, together with the reference non-accelerated experiments. Surprisingly, even if they both start with the same initial ice sheet geometry and spun-up climate, the accelerated experiment (black line) displays a drastically different time evolution compared to its non-accelerated counterpart (pink line). In fact, even without accounting for the freshwater feedback, the accelerated experiment presents a collapse in the AMOC in the course of TII, while this collapse is absent in the non-accelerated experiment. This difference between accelerated and non-accelerated AMOC changes was not observed for TI (Quiquet et al., 2021). This means that, independently from the freshwater flux, the oceanic

circulation across TII seems more unstable in our model. The collapse of the AMOC in the accelerated experiments happens nonetheless later than when the freshwater flux is accounted for. It occurs at the time of minimal AMOC strength in the non-accelerated experiments, circa 129 kaBP.

395

Fig. 13 also displays the effect of changing the initial ice sheet geometry. These alternative ice sheets are presented in Fig. 12. They consist in a slightly reduce North American ice sheet (-6 % in volume) and a larger Eurasian ice sheet towards the East and the South. A slightly larger Eurasian ice sheet volume (+36 %) has a negligible impact on the large scale climate evolution through TII since global mean temperature, AMOC and sea ice changes remain very similar in this case with respect to the reference experiments (Fig. 13). This is not necessarily surprising since this sensitivity experiment presents no change in global ice mass stored on land but only a slight geographical distribution change. It is only with a substantially larger (+71 %) Eurasian ice sheet volume that we can observe significant changes in the simulated climate. Since this simulation presents an increase in the amount of land ice with respect to the reference experiment, it shows a decrease in global mean temperature of about 0.3 °C through the termination (Fig. 13a). The AMOC collapse is also delayed by about 500 years with respect to the reference experiment. In all these accelerated simulations, the AMOC abruptly recovers towards the end of the last interglacial period. However, the timing of the recovery is impacted by the choice of the initial ice sheet geometry: the AMOC recovers almost 2000 years (resp. 500 years) earlier than the reference experiment when starting from a substantially larger (resp. slightly larger) Eurasian ice sheet. This highlights the long timescales, greater than 5000 years, at play for the coupled ice sheet – climate model. Nonetheless, the initial ice sheet geometry seems overall to play a secondary role in the climate evolution across TII. In addition to its effects on climate, the initial ice sheet configuration can impact the evolution of ice sheet volume in the course of TII (Fig. S7). When using the slightly larger Eurasian ice sheet and slightly smaller North American ice sheet there is no change in the total ice volume evolution with respect to the standard version. This suggests that changing moderately the land ice distribution with no change in total volume does not impact the total ice sheet retreat. This is different when looking at the total ice volume evolution using the much larger Eurasian ice sheet initial condition. In this case the deglaciation of all the Northern Hemisphere ice sheets tends to be delayed. Although initially smaller compared to our reference configuration, the North American ice sheet retreats almost one thousand years later when using the largest Eurasian ice sheet. This is mostly due to the fact that the larger extent of the Eurasian ice sheet in this case produces a high-latitude cooling, especially in summer (Fig. S8).

We also used the accelerated experiments to assess the respective role of external forcings (orbital configuration and green-house gas concentration) and internal feedbacks (ice sheets and vegetation). In these experiments one of these aspects is fixed at its initial value while the rest of the system is free to evolve following the standard external forcings. The results of these experiments for TII and TI in terms of the global mean temperature is shown in Fig.14. The two external forcings, greenhouse gas concentration (GHG) and orbital configuration (ORB), are equally important. Discarding one or the other results in much lower temperatures during the Holocene or at the peak warmth during the last interglacial. Interestingly, for the second half of the TII experiment they induce opposite trends: warming for fixed orbit and cooling for fixed greenhouse gas concentration. Ice sheet changes (ICE), the major internal feedback, produce an impact as large as the two external forcings. This means

that the ice sheet-climate feedback is particularly strong in the model as it explains half the glacial-interglacial temperature change. The vegetation feedback (VEG) has a smaller impact on the global mean temperature since it is the closest to the reference experiments (ALL). However, discarding the vegetation change leads to an underestimation of the glacial-interglacial temperature change of about 1°C. The predominant effect for the vegetation feedback is that keeping a glacial vegetation cover tends to produce higher surface albedo. For these sensitivity experiments, the changes in terms of temperature are somehow hiding ice sheet changes, presented in Fig. 15. While the orbital configuration and the greenhouse gas concentration were both considered as equally important for the temperature, the deglaciation of the ice sheets is primarily caused by the change in the orbital configuration. In fact, for TII, the fixed greenhouse gas concentration experiment (GHG) produces an ice volume very close to the reference experiment (ALL), only delaying slightly the ice loss. The role of this forcing is even smaller than the vegetation feedback (VEG) to explain the Northern Hemisphere ice sheet retreat. This relative importance of orbital configuration, greenhouse gas concentration and vegetation is mostly shared among the two terminations except for the early part of TI. For this period, the reference experiment shows a slight increase in ice volume which is only explained by the combination of the two external forcings, which display a very moderate reduction (Fig. 1). However, as for TII, the ice sheet retreat for TI is primarily due to insolation changes. If the predominant role of insolation to explain the ice sheet retreat was already identified by others (e.g. Ganopolski and Calov, 2011), it might also be amplified in our case by the relatively low climate sensitivity of our model (about 2°C, Loutre et al., 2011).

## 4 Discussion

In our reference TII experiments we made a critical assumption by using the LGM ice sheets as initial ice sheet geometry. There were two main motivations for this choice. First, in doing so, it is easier to compare the two glacial terminations in terms of timing of deglaciation and large scale climatic signals. Second, it is very challenging to properly initialise a coupled ice sheet – climate model at the PGM given the lack of strong constraints on ice sheet geometry at this time. There are nonetheless several lines of evidence that suggest a smaller North American ice sheet and larger Eurasian ice sheet at the PGM with respect to the LGM (Svendsen et al., 2004; Lambeck et al., 2006; Colleoni et al., 2016; Batchelor et al., 2019; Pollard et al., 2023). This fundamental difference between the PGM and LGM ice sheet geometries can potentially have a large impact on atmospheric circulation and, in fine, subsequent ice sheet dynamics. To have an idea of the implication of our model simplification, we show in Fig. 16 the change in the winter and summer atmospheric circulation between the PGM and the LGM. When using the same ice sheets for the two glacial maximums, as in our reference experiments, the change in external forcings (orbital configuration and greenhouse gas concentration) leads to changes in the atmospheric circulation, especially during boreal summer, when the insolation difference is the largest (Fig. 16b,d). In this case, there is a slight weakening of the summer North Atlantic anticyclonic and Siberian cyclonic circulations. These moderate changes in summer circulation are amplified when using a substantially larger (+71 % ice volume) Eurasian ice sheet (Fig. 16c,e). However, the major difference is in winter: while there was no major difference in atmospheric circulation between the LGM and the PGM with the reference ice sheets, the larger

Eurasian ice sheet leads to a much stronger (respectively weaker) anticyclonic pattern in North America (respectively Siberia). These changes in circulation when using a larger Eurasian ice sheet leads to an increase in winter precipitation in Eurasia and a decrease in North America (Fig. S9). This result is somewhat symmetrical to the one of Beghin et al. (2015) who showed that the topographic effect of the North American ice sheet reduces the precipitation in Eurasia through planetary wave changes. It is also consistent with Liakka et al. (2016) who suggested that the development of a large Eurasian ice sheet in its eastern part is favoured by smaller than LGM North American ice sheet.

The simulated atmospheric circulation changes when using different ice sheet geometries at the PGM do not seem to impact drastically the individual ice sheet volume evolution through TII (Fig. S7). These can be caused by the low spatial resolution of our atmospheric model that can underestimate the atmospheric circulation changes. For example, Lofverstrom and Liakka (2018) used an atmospheric-only general circulation model at various spatial resolutions to generate climate forcings to run stand-alone ice sheet model simulations. They showed that the model ability to reconstruct the LGM ice sheets strongly depends on the spatial resolution of the atmospheric model, higher resolution showing generally better performance. The authors suggest in particular that the T21 spatial resolution is fundamentally inadequate to resolve numerically the baroclinic waves. Indeed, to insure stability of the numerical scheme, coarse resolution models show a larger diffusivity which dampens the waves (Magnusdottir and Haynes, 1999; Polvani et al., 2004; Lofverstrom and Liakka, 2018). However, while we use a T21 resolution, our model temperature biases are not comparable to the ones shown in Lofverstrom and Liakka (2018). For example, they show that their model at T21 is unable to reconstruct the Eurasian ice sheet, independently from the surface mass balance scheme they use. In our case, the model does build up an ice sheet in Western Eurasia and none in Siberia, even without the indirect bias correction that we use in the melt equation (Eq. 2 leads to increase $c_{rad}$ in Eurasia, inducing more melt). This suggests that other biases (apart from numerical diffusion) can alter model performance and that the fact that our model correctly represents the LGM ice sheets might be the results of some compensating biases. More generally, using outputs from the Paleoclimate Modelling Intercomparison Project (PMIP) phase 3 and 4 LGM database to force ice sheet models, both Niu et al. (2019) and van Aalderen et al. (2023) show that most general circulation models do not provide suitable climatic forcing fields to reconstruct ice sheets in agreement with geological reconstructions. These deficiencies are generally not related to spatial resolution differences amongst participating models. However, for a given climate model, a higher spatial resolution will tend to have a more accurate representation of the topography and this will induce noticeable difference with its lower spatial resolution version (Lohmann et al., 2021). In fact, SMB is highly correlated to topography, notably due to the direct impact of elevation on surface temperature. This is why different groups have used different strategies to downscale ice-processes (Robinson et al., 2010; Fyke et al., 2011; Krebs-Kanzow et al., 2021; Crow et al., 2024). While the downscaling scheme that we use does not allow any improvement in the topographically-induced atmospheric circulation change, it nonetheless better capture the melt elevation feedback than a standard vertical lapse rate approach.

Apart from atmospheric model resolution, other simplifications in our climate model can have an impact on the simulated ice sheet and climate trajectories through the terminations, such as for example the simple vegetation or surface mass balance

schemes. Unfortunately, there are not many modelling studies that have simulated the TII terminations with a coupled ice sheet – climate model to compare our model results to. Using the CLIMBER-2 model, Ganopolski and Brovkin (2017) also produce an early collapse of the AMOC during TII circa 132 kaBP but they do not focus particularly on the TII with respect to TI. More studies have focused on the question of the last interglacial sea level. For example, Goelzer et al. (2016) use LOVECLIM to simulate the evolution of the Greenland and Antarctic ice sheets during the last interglacial (135-115 kaBP). In their work they impose the geometry of the other Northern Hemisphere ice sheets assuming a consistent deglaciation pattern between TII and TI. They produce higher sea level contribution from the Greenland ice sheet compared to our experiments but this is most likely due to difference in surface mass balance computations. While we use the absolute climate forcing computed by iLOVECLIM, they use an anomaly method superimposed to a reference modern climate. They also use a scaling factor for temperature to account for the low sensitivity of LOVECLIM. Another relevant study is the one of Sommers et al. (2021) who used a general circulation model coupled to a Greenland ice sheet model to simulate ice sheet and climate evolution through the last interglacial (127-119 kaBP). A direct comparison with our work is not necessarily trivial, since our main target is different. Sommers et al. (2021) investigate the Greenland ice sheet response to the peak insolation while our goal is to investigate the differences and similarities between the last two deglaciations of the Northern Hemisphere ice sheet. As a result, for example, Sommers et al. (2021) start from a climate equilibrium under 127 kaBP boundary condition which likely biased their initial climate towards higher temperature since 127 kaBP is close to the Northern Hemisphere summer insolation maximum. Nonetheless, their major finding is that the vegetation feedback plays a major role in the magnitude of Greenland mass loss. This is consistent with our results since the minimum Greenland ice sheet volume is 20 % larger when using a constant glacial vegetation instead of the interactive vegetation (Fig. 15a).

## 5  Conclusion

In this paper we have presented modelling experiments of the last two glacial terminations using a coupled climate – ice sheet model. We have shown that the two terminations display a number of important similarities. Notably, while the strength of the overturning Atlantic circulation is similar for the last and penultimate glacial maximum, freshwater flux can lead to its complete and irreversible shutdown for the two terminations. The ice geometries through the two terminations are also very similar. This means that, in our model, changes in external forcings alone are not able to explain different ice sheet configurations through the terminations if the glacial configurations are the same. For the two terminations, insolation is the main driver for ice sheet retreat while greenhouse gas concentration has only a minor role. However, the predominant role of insolation might also be the result of the relatively low climate sensitivity of our model. Beyond these similarities, the two terminations display also important differences, primarily caused by differing insolation evolution. TII presents a more rapid Northern Hemisphere warming and ice sheet melt relative to TI which explains the higher ice sheet contribution to sea level rise during the last interglacial period compared to the Holocene. However, in the Southern Hemisphere, the weaker insolation leads to lower SST through TII, persisting into the last interglacial period, in disagreement with proxy-based reconstructions. Southern Ocean sub-surface

temperature are nonetheless higher during TII, which can be consistent with a more retreated Antarctic ice sheet during the last interglacial period, not simulated as part of our setup. Finally, while the AMOC is prone to collapse for both terminations,

this sensitivity is much larger for TII where a collapse without freshwater flux is simulated in some experiments. This suggests that, apart from freshwater flux, external forcing differences among the two terminations can induce different AMOC evolution.

*Data availability.* Archiving of source data of the figures presented in the main text of the manuscript is underway. Data will be made publicly available upon publication of the manuscript on the Zenodo repository with digital object identifier 10.xxxx/zenodo.xxxxxxx. They

are temporarily available for review purposes upon request.

*Author contributions.* A.Q. designed the project and performed the simulations. A.Q. and D.M.R. have contributed to the model developments necessary to perform this work. A.Q. and D.M.R. participated in the analysis of model outputs and the manuscript writing.

*Competing interests.* The authors declare no competing interests.

*Acknowledgements.* We acknowledge the Institut Pierre Simon Laplace for hosting the iLOVECLIM model code under the LUDUS frame-

540 work project (https://forge.ipsl.jussieu.fr/ludus). This work was supported by ANR PIA funding: ANR-20-IDEES-0002. It also received funding from the French National Research Agency under Grant ANR-19-CE01-0015 (EIS).

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

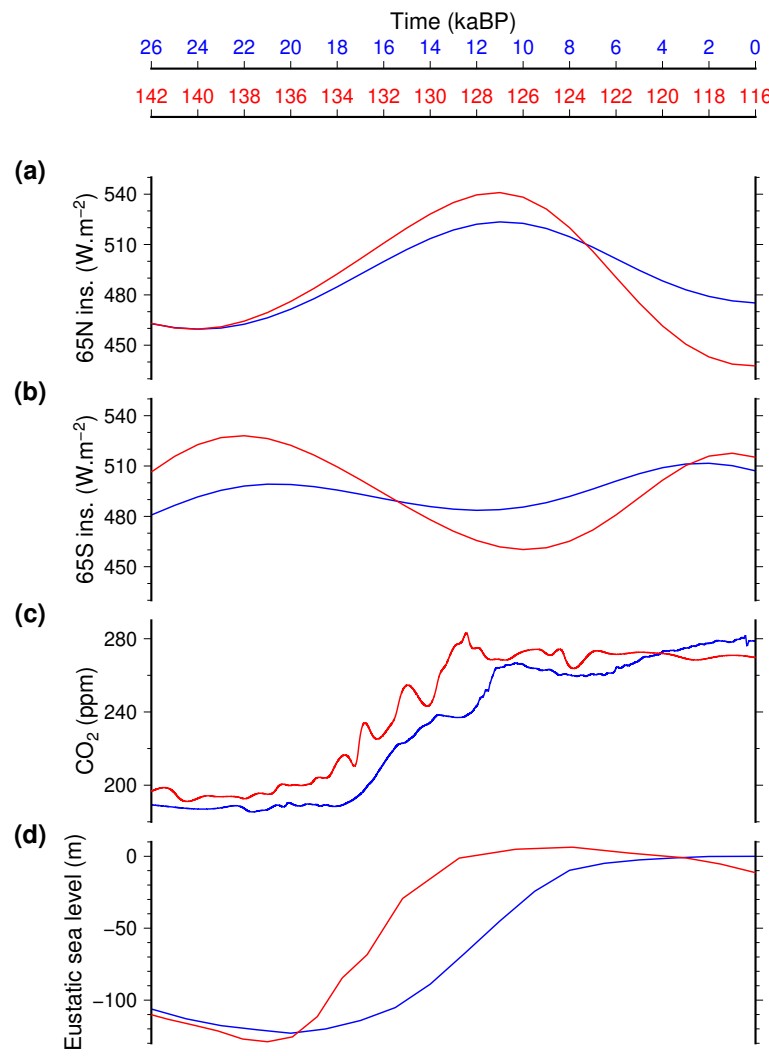

**Figure 1.** Temporal evolution of the major forcings over TII (red) and TI (blue). **(a)**: June mean insolation at 65°N and **(b)**: december mean insolation at 65°S (Berger, 1978). **(c)**: carbon dioxide mixing ratio (Lüthi et al., 2008). **(d)**: eustatic sea level (Waelbroeck et al., 2002).

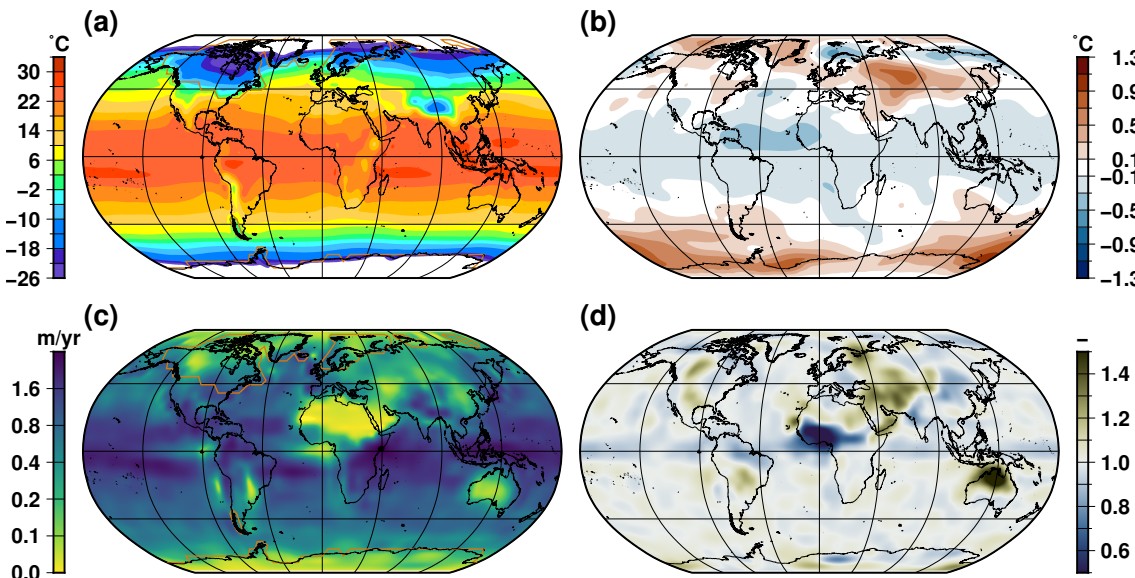

**Figure 2.** Temperature and precipitation for glacial initial conditions. (**a**): climatological annual mean near surface air temperature computed at 26 kaBP. (**b**): 142 kaBP temperature difference with respect to 26 kaBP. (**c**): climatological annual mean precipitation at 26 kaBP. (**d**): 142 kaBP precipitation ratio relative to 26 kaBP. The orange line is the extent of the ice sheets.

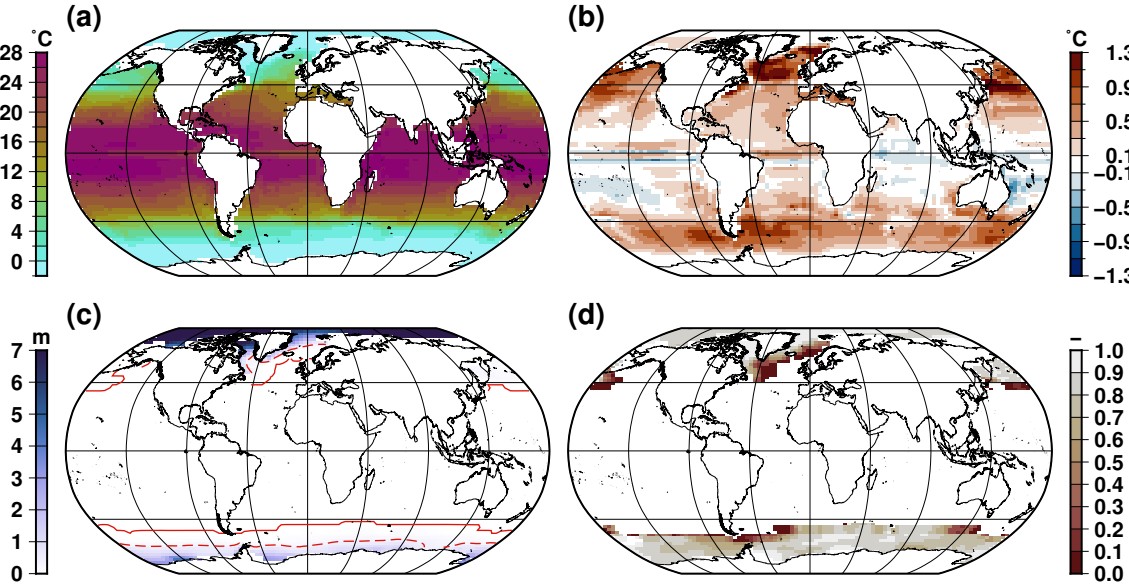

**Figure 3.** Sea surface temperature and sea ice thickness for glacial initial conditions. (**a**): climatological annual mean sea surface temperature computed at 26 kaBP. (**b**): 142 kaBP temperature difference with respect to 26 kaBP. (**c**): climatological annual mean sea ice thickness at 26 kaBP. The continuous red line stands for the maximal sea ice extent at 26 kaBP and the dashed line stands for a mean annual thickness of 0.5 cm at 26 kaBP. (**d**): 142 kaBP thickness ratio relative to 26 kaBP.

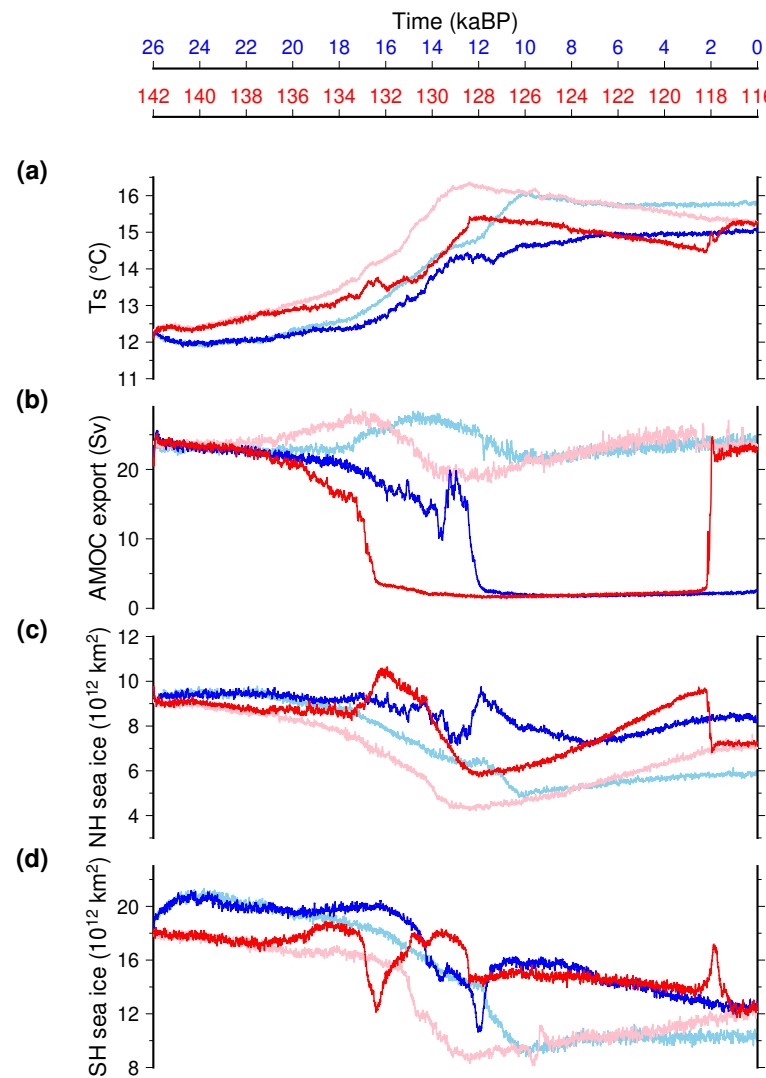

**Figure 4.** Temporal evolution of large scale climate features across TII (red) and TI (blue). **(a)**: Simulated global mean surface temperature. **(b)**: Simulated maximum of the Atlantic stream function. **(c)**: Northern Hemisphere sea ice extent. **(d)**: Southern Hemisphere sea ice extent. Here, we use a 20-yr running mean for the model results to smooth interannual variability. Light colors are the experiments that do not account for the freshwater water feedback from ice sheet melting.

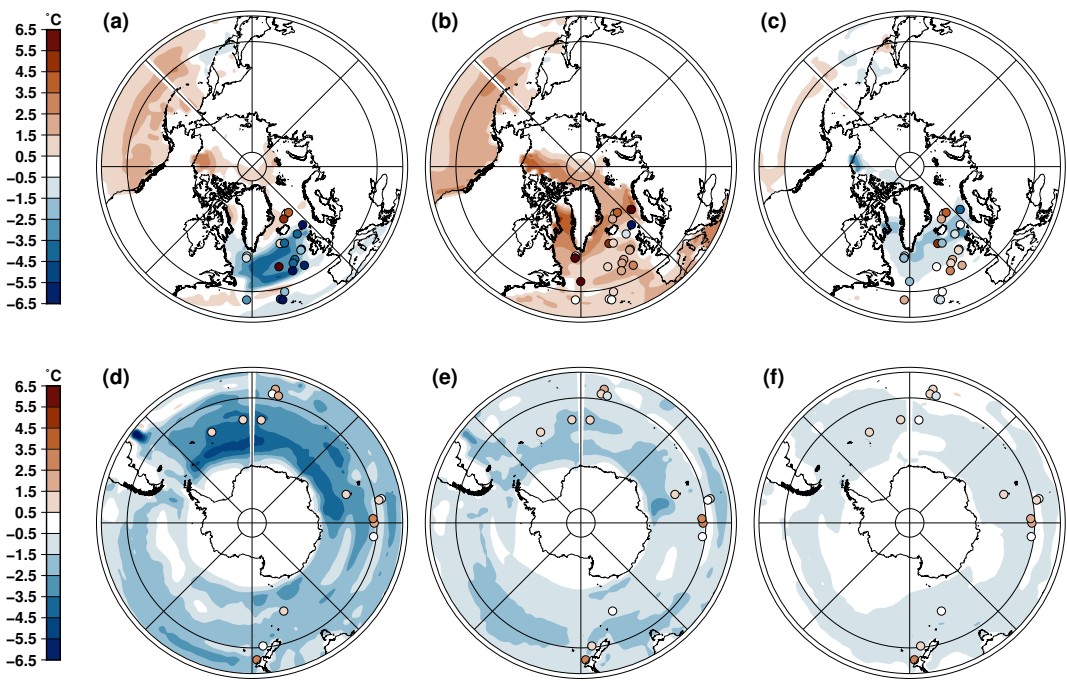

**Figure 5.** Last interglacial summer sea surface temperature anomalies with respect to the pre-industrial (0 kaBP). High latitudes anomalies in the Northern Hemisphere (respectively Southern Hemisphere) at 130 kaBP **(a)** (resp. **(d)**), 125 kaBP **(b)** (resp. **(e)**) and 120 kaBP **(c)** (resp. **(f)**). Proxy based reconstructions from Capron et al. (2017) are shown in circles. Summers are defined as the warmest three months. Anomalies are computed with the experiment that account for the freshwater flux feedback resulting from ice sheet melting.

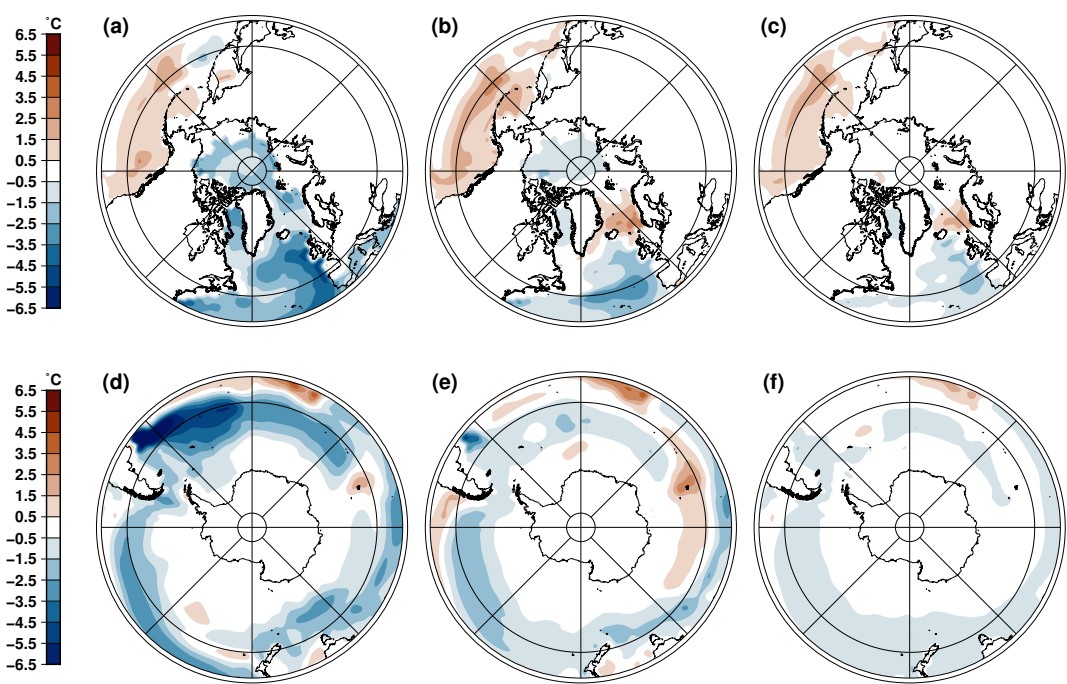

**Figure 6.** Last interglacial annual temperature anomalies with respect to the pre-industrial (0 kaBP) at 220 m depth. High latitudes anomalies in the Northern Hemisphere (respectively Southern Hemisphere) at 130 kaBP **(a)** (resp. **(d)**), 125 kaBP **(b)** (resp. **(e)**) and 120 kaBP **(c)** (resp. **(f)**). Anomalies are computed with the experiments that include the freshwater flux feedback resulting from ice sheet melting.

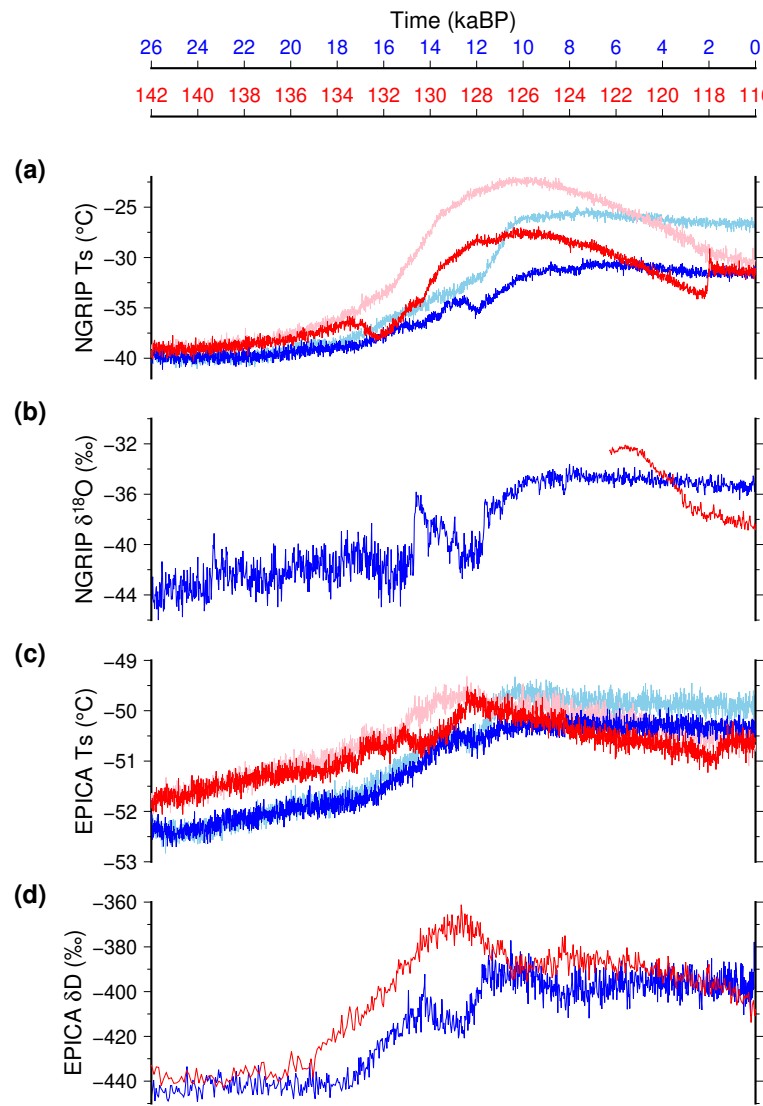

**Figure 7.** Simulated temperature and temperature proxy over Greenland and Antarctica across TII (red) and TI (blue). **(a)**: Simulated temperature and **(b)**: $\delta_{18}O$ (Andersen et al., 2004; Lemieux-Dudon et al., 2010) at North GRIP. **(c)**: Simulated temperature and **(d)**: deuterium excess (Jouzel et al., 2007; Lemieux-Dudon et al., 2010) at EPICA DOME C. For the model results, we use a 20-yr running mean for the model results to smooth interannual variability. Light colors are the experiments that do not account for the freshwater water feedback from ice sheet melting.

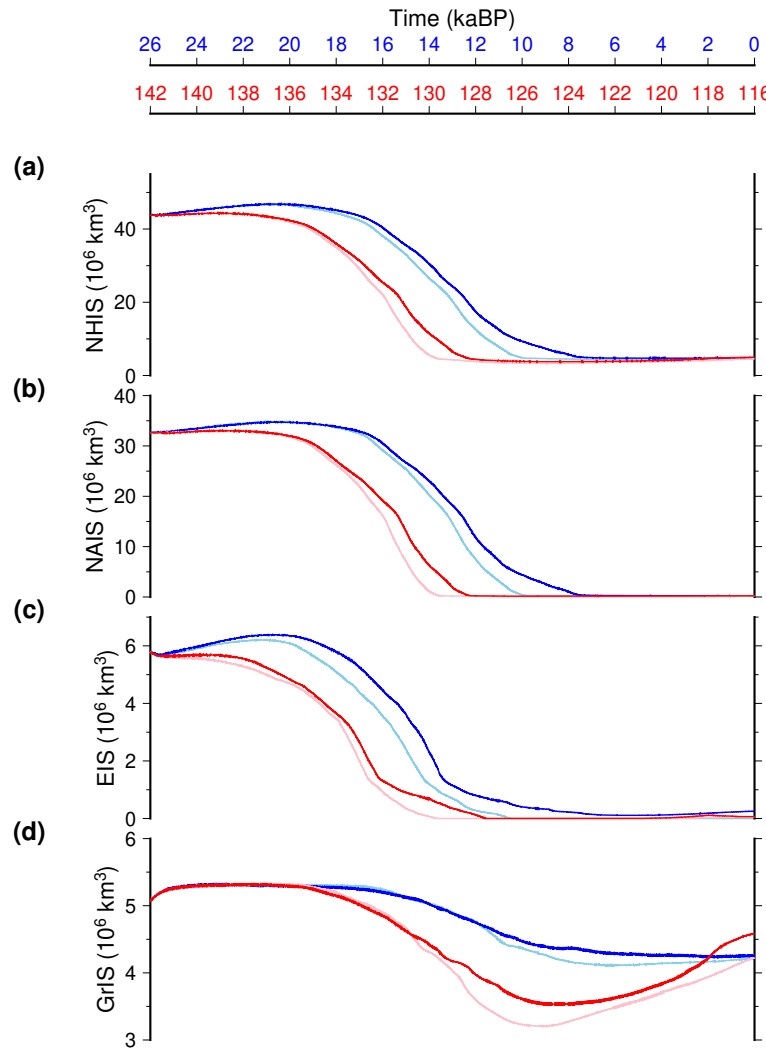

**Figure 8.** Temporal evolution of individual ice sheet total ice volume across TII (red) and TI (blue). **(a)**: total North Hemisphere ice sheet volume. **(b)**: North American ice sheet volume. **(c)**: Eurasian ice sheet volume. **(d)**: Greenland ice sheet volume. Light colors are the experiments that do not account for the freshwater water feedback from ice sheet melting.

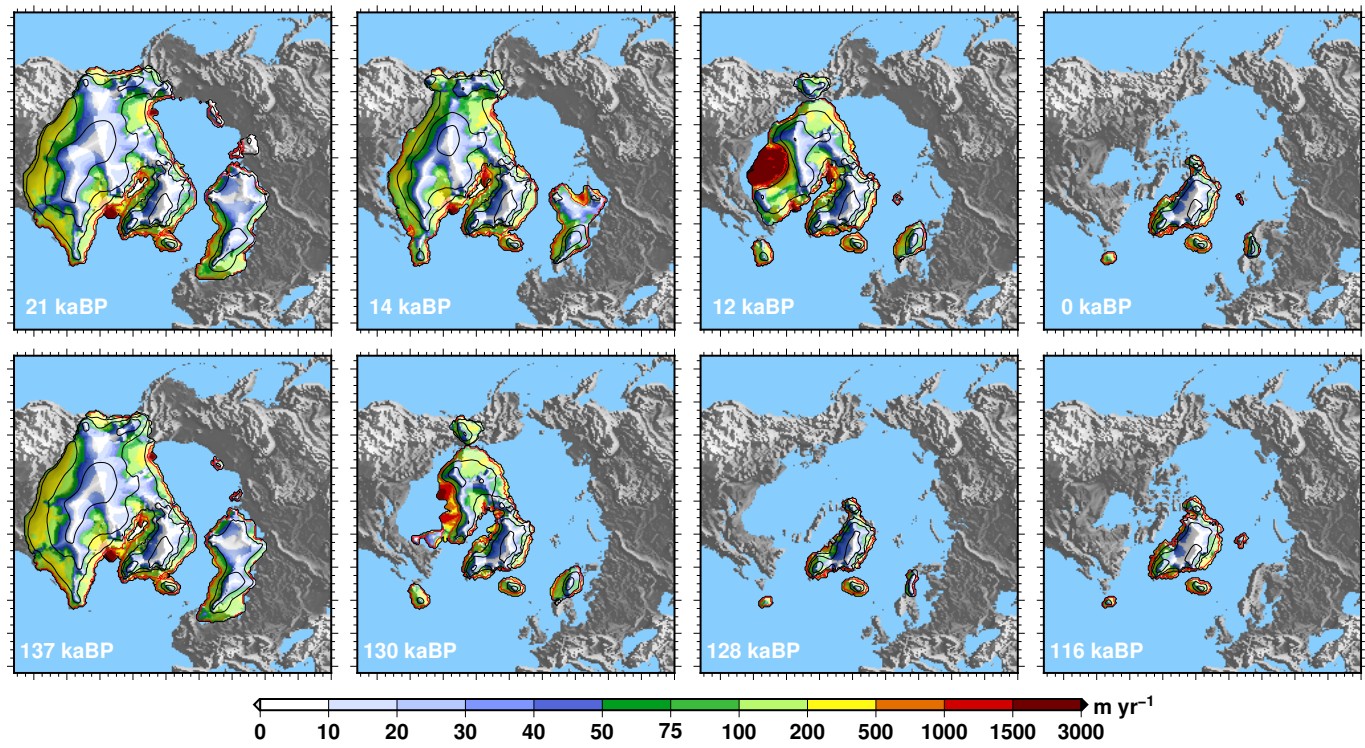

**Figure 9.** Simulated Northern Hemisphere ice sheets across the two terminations. Four selected snapshots are shown for TI (top) and TII (bottom). The dates of the snapshots are chosen to be at 5, 12, 14 and 26 ka after the start of the experiments for the two terminations. The black isocontours show the simulated ice elevation above contemporaneous eustatic sea level (contours separated by 1000 metres). The red contour is the ice sheet grounding line. The colour palette represent the amplitude of the simulated vertically averaged ice sheet velocity, draped over the surface topography. The experiments shown here include the freshwater flux feedback resulting from ice sheet melting.

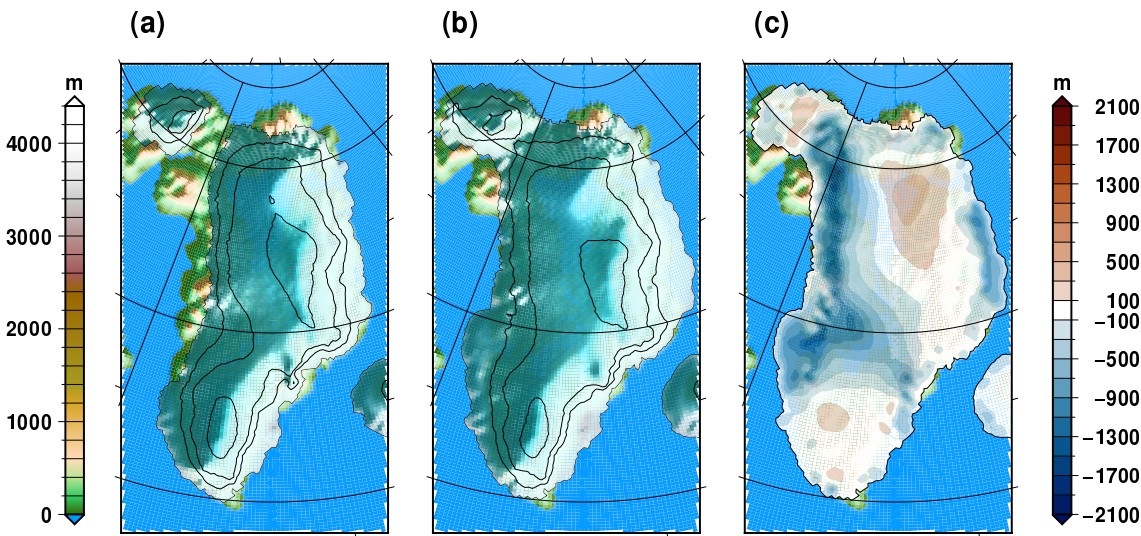

**Figure 10.** Simulated Greenland ice sheet topography. **(a)**: at 125 kaBP, minimum of the TII GrIS volume. **(b)**: at 0 kaBP, the end of the TI experiment. **(c)**: Ice thickness difference (a-b). In (a) and (b) the black contours represent iso-elevations every 1000 m for the glaciated regions. The experiments shown here include the freshwater flux feedback resulting from ice sheet melting.

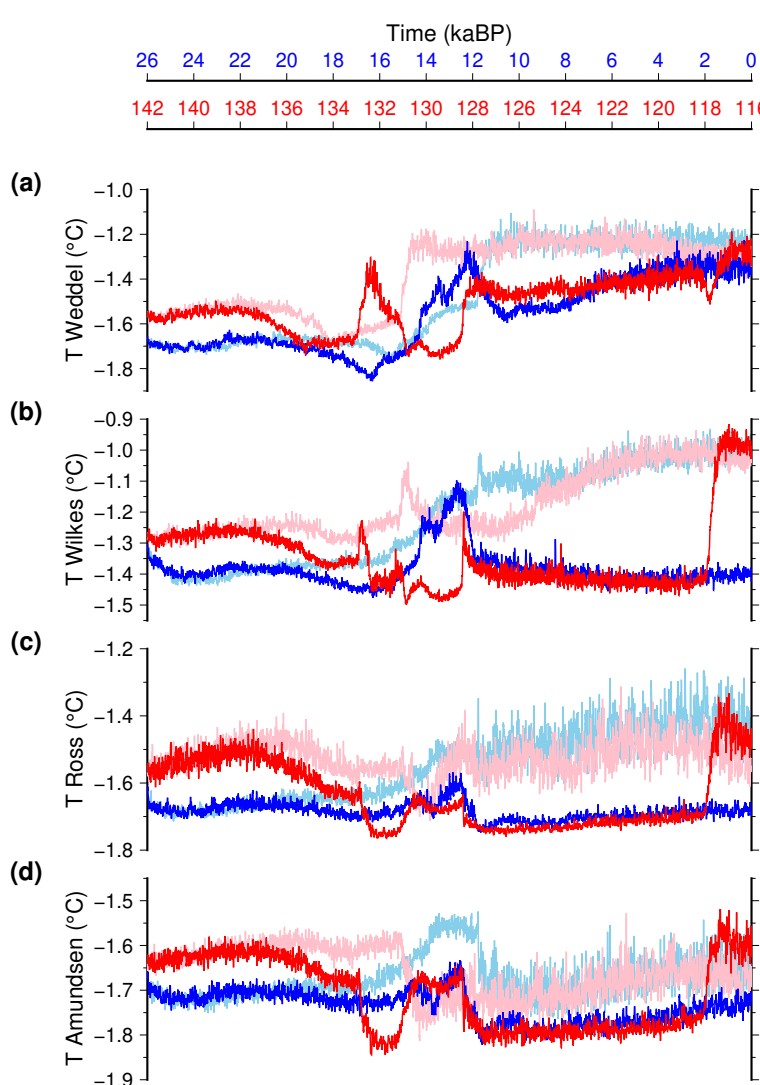

**Figure 11.** Temporal evolution of Southern Ocean sub-surface (600 m) temperature across TII (red) and TI (blue). **(a)**: Weddel sea, averaged over longitudes ranging from 300°E to 340°E and latitudes from -90°N to -70°N. **(b)**: Wilkes sector, averaged over longitudes ranging from 124°E to 170°E and latitudes from -90°N to -64°N. **(c)**: Ross sea, averaged over longitudes ranging from 183°E to 207°E and latitudes from -90°N to -72°N. **(d)**: Amundsen sea, averaged over longitudes ranging from 245°E and 260°E and latitudes from -90°N to -68°N. We use a 20-yr running mean for the model results to smooth interannual variability. Light colors are the experiments that do not account for the freshwater water feedback from ice sheet melting.

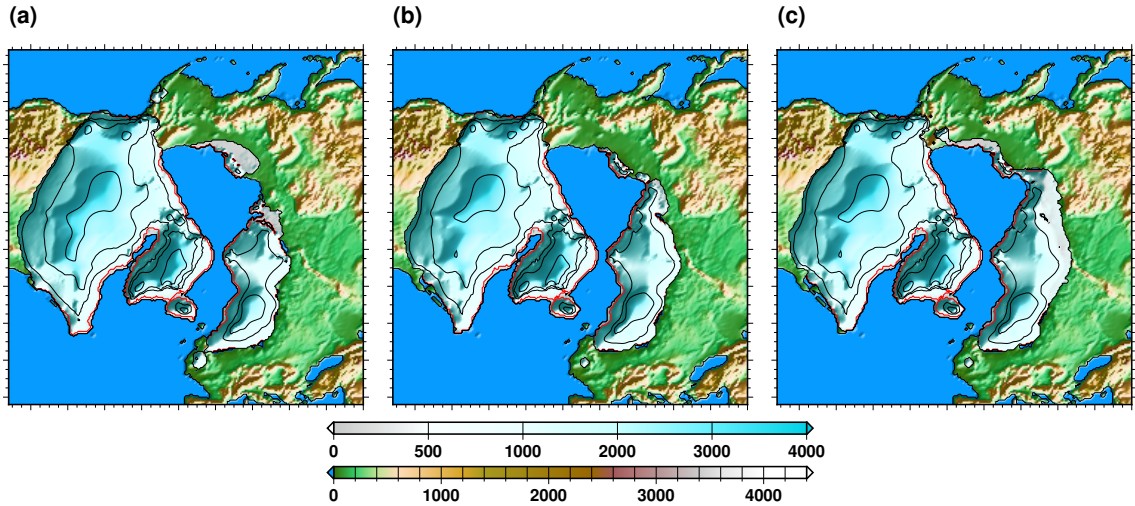

**Figure 12.** Initial ice sheet topographies for the sensitivity experiments. **(a)**: Reference ice sheet. **(b)**: Slightly reduced North American ice sheet (-8 % in ice volume) and larger Eurasian ice sheet (+36 %). **(c)**: Slightly reduced North American ice sheet (-6 %) and much larger Eurasian ice sheet (+71 %).

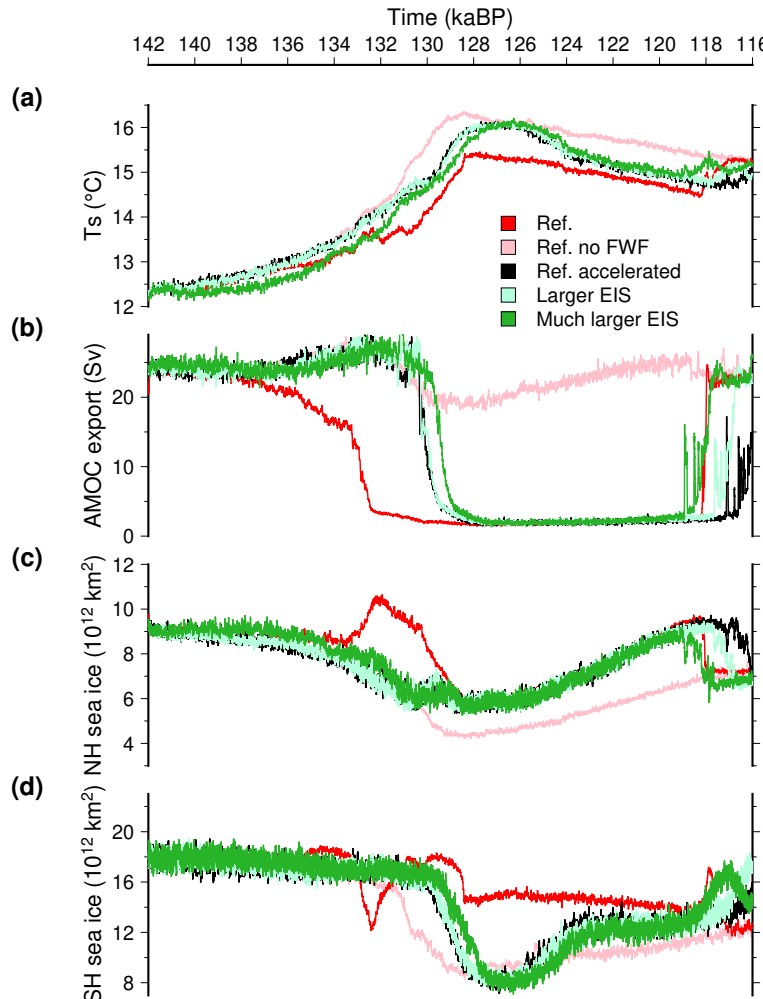

**Figure 13.** Temporal evolution of large scale climate features across TII for asynchronously coupled experiments. **(a)**: Simulated global mean surface temperature. **(b)**: Simulated maximum of the Atlantic stream function. **(c)**: Northern Hemisphere sea ice extent. **(d)**: Southern Hemisphere sea ice extent. Here, we use a 10-yr running mean for the model results to smooth interannual variability. The synchronously reference experiments with and without the freshwater flux feedback are shown in red and pink, respectively, as in Fig. 4. The accelerated experiment that uses the reference ice sheet is in black while the experiments with slightly (+36 %) and substantially larger (+71 %) Eurasian ice sheet volume are in light and dark green, respectively.

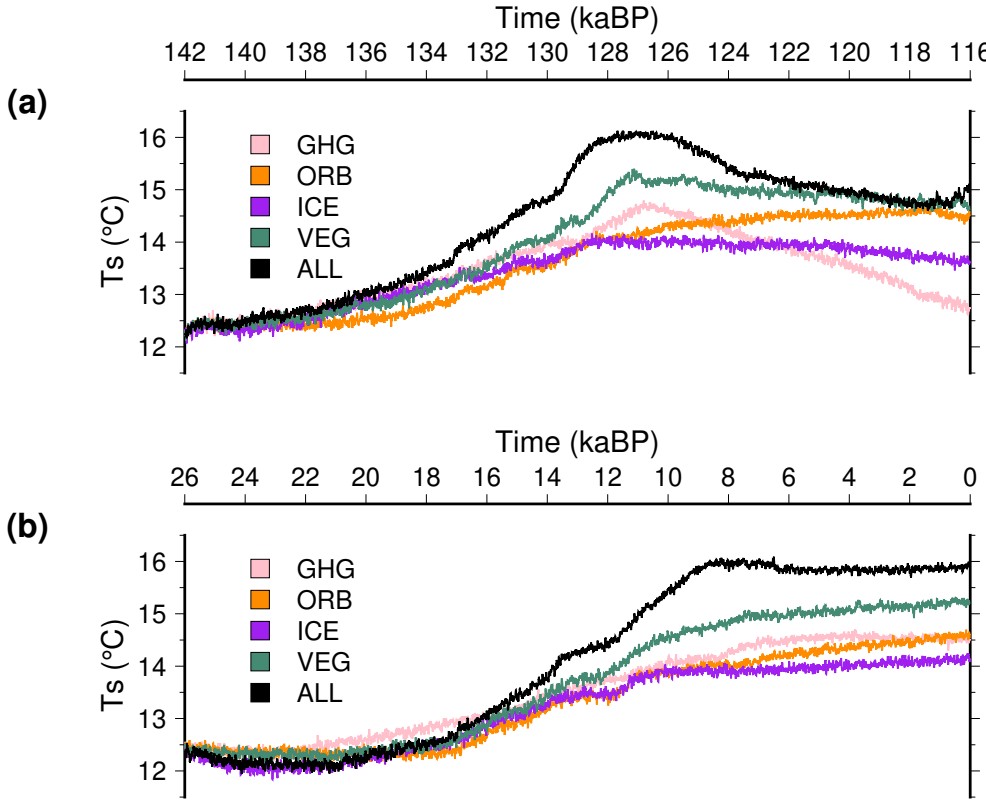

**Figure 14.** Temporal evolution of the global mean surface temperature for the experiments with constant greenhouse gas concentration (GHG), with constant orbital parameters (ORB), with a fixed ice sheet mask and orography (ICE) and with a fixed vegetation (VEG). **(a)**: For TII. **(b)**: For TI.

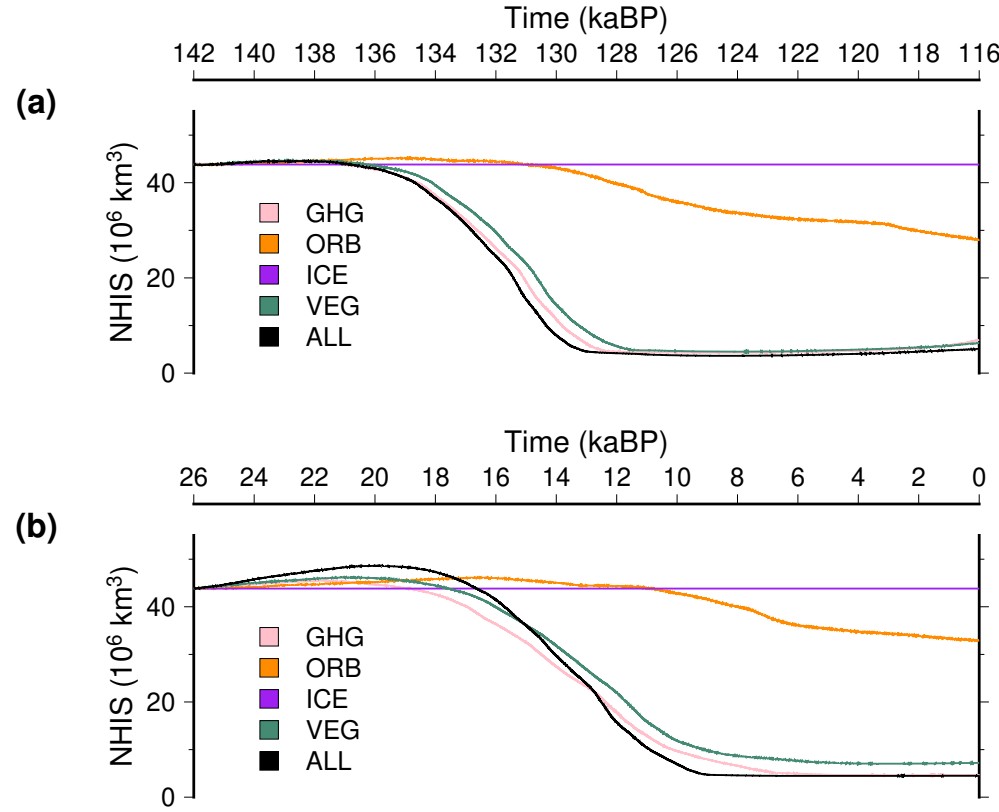

**Figure 15.** Temporal evolution of the total Northern Hemisphere ice volume for the experiments with constant greenhouse gas concentration (GHG), with constant orbital parameters (ORB), with a fixed ice sheet mask and orography (ICE) and with a fixed vegetation (VEG). **(a)**: For TII. **(b)**: For TI.

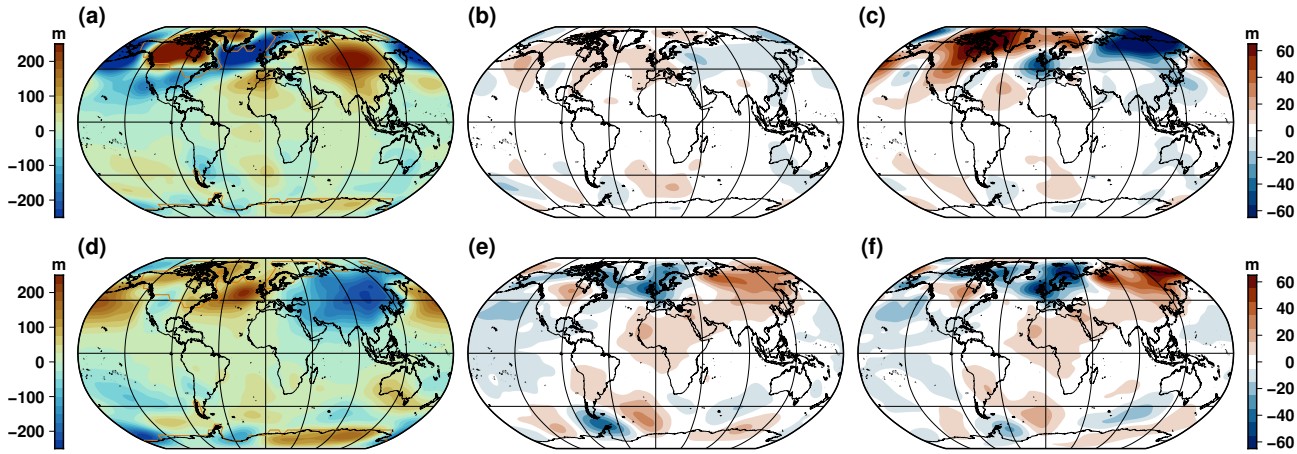

**Figure 16.** Simulated change in the atmospheric circulation at the PGM. **(a)**: Winter DJF anomaly of the geopotential height (800 hPa) with respect to its zonal mean at the LGM. **(b)**: Difference of this geopotential height anomaly at the PGM with respect to the LGM in winter in the reference experiment. **(c)**: Same difference but using the substantially larger (+71 %) Eurasian ice sheet as boundary condition. **(d)**, **(e)**, **(f)**: Same as **(a)**, **(b)**, **(c)** but for summer JJA.