# Peer review of "Investigating similarities and differences of the penultimate and last glacial terminations with a coupled ice sheet – climate model"

_EGUsphere, 2023_

## Author Comment (AC1)

In the following, the reviewer comment are in blue and our response in black.

The manuscript presents the results of a fully coupled Northern Hemisphere ice sheet—climate model applied to the last two glacial terminations. The manuscript is well-written and nicely illustrated. The description of the model, coupling and sensitivity analysis is mostly clear but could benefit from some minor additions. Overall, I enjoyed reading this paper and I am sympathetic to the aims. I am not suggesting the authors conduct additional experiments. I hope my comments help in improving the manuscript.

Thanks for your your positive evaluation of our manuscript and your useful comments. We have taken them into account for our revised version. Detailed answer to your individual comments are provided in the following.

Comments

Alternative PGM ice sheet geometry:

The manuscript would benefit from a more detailed explanation of how the alternative ice sheet geometry has been applied. In the methods section, it is only briefly mentioned in L169 and in the results (L345 to L348). It would be valuable to explore the regional and large-scale impacts on the climate resulting from this new ice sheet configuration as well as its implications on the timing and on the deglaciation history during the TII.

We agree that we did not put too much weight on these sensitivity experiments in the initial version of our manuscript. The alternative ice sheets have been obtained by changing regionally the ablation parameters during the ice sheet spin-up (uncoupled experiments). These parameters were increased in North America (more melt) and reduced in Eurasia (less melt). Then we simply used these new ice sheets as initial ice sheet conditions for our transient alternative TII experiments. We added a few sentences in the revised manuscript to make it clearer.

We have also included a figure that present the individual ice sheet volume evolution (Fig. RA1) through TII for the three initial ice sheet states. As shown in this figure, there is no major differences between these experiments using alternative geometries and the reference experiment. The largest Eurasian ice sheet helps maintaining a colder Northern Hemisphere climate. This tends to delay the retreat of all the Northern Hemisphere ice sheets, including the North American one. Although initially smaller compared to our reference configuration, the North American ice sheet retreats almost one thousand years later when using the largest Eurasian ice sheet. These additional elements of discussions have been added in the revised manuscript.

The revised manuscript now contains a discussion section in which we present the atmospheric circulation differences between the PGM and the LGM, focusing on the impact of these different ice sheet topographies.

[Figure]

**Figure RA1.** Temporal evolution of individual ice sheet total ice volume across TII using different initial ice sheet geometries. **(a)**: Total North Hemisphere ice sheet volume. **(b)**: North American ice sheet volume. **(c)**: Eurasian ice sheet volume. **(d)**: Greenland ice sheet volume. The experiment that uses the reference ice sheet is in black while the experiments with slightly larger (+36 %) and larger (+71 %) Eurasian ice sheet volume are in light and dark green, respectively.

Atmospheric resolution:

It would be beneficial to include a discussion on the limitations due to the climate resolution. The simulations are based on the intermediate complexity climate model iLOVECLIM, with an atmospheric resolution of T21. Previous studies have established the implications of coarseresolution climate models in the modelling during the last glacial maximum and the deglaciation (eg. Lofverstrom et al., 2018; Lohmann et al., 2021).

It is true that the atmosphere in iLOVECLIM is simplified. The dynamical core uses the quasi-geostrophic approximation with some additional ageostrophic terms for a better representation of the Tropical circulation, in particular Hadley cells (Opsteegh et al., 1998). We agree that the atmospheric model resolution, but also simplification in its physics, can have important impact on the simulated ice sheets. This is now discussed, also with respect to the existing literature, in the new discussion section of the revised manuscript.

Other concerns:

To make the paper more accessible to a broader audience, including non-modellers, it may be helpful to explicitly state that the primary aim is not to precisely replicate the timing and pattern of deglaciation but rather to explore the model's sensitivity throughout both terminations. This clarification can aid in ensuring that readers from various backgrounds can appreciate the study's objectives and outcomes.

We have added the following towards the end of the introduction:
"Using a relatively simplified setup, we do not aim to precisely match the available proxy data but instead we aim at better understanding the role of external forcings (orbital configuration and greenhouse gas concentration) on glacial terminations."

Technical comments:

L231. "In ?"

Sorry for this, it should have been "In Quiquet et al. (2021)". Corrected.

L245. its written "kyrs" while in some other parts of the text is written "kyr" (eg. L283). Moreover, in other parts is written "ka" (L292). Please check.

Thanks for pointing these inconsistencies. We now use "ka" for durations and "kaBP" for dates.

Figs. 1 - 13. It is written "kaBP" while in Figs 14 and 15 "ka BP".

We have changed Fig. 14 and 15 to be consistent with the rest of the paper.

Fig 7. Keep the design of the other figures

Changed, we have put all the y-axis on the left-hand side of the figure.

Fig 12. Replace "rhe" for "the"

Corrected.

Fig. 13. Include legend

Done.

Fig 14 and 15. Keep the design of the other figures

In the rest of the paper, the two terminations are shown in the same panel using two different colours. It is true that here we have separated the two terminations in two distinct panels. The reason is that we have to show 5 different experiments (ALL, ORB, GHG, ICE and VEG) for the two terminations. Grouping all this information in one panel would have made the results difficult to read. We have kept our representation with two panels but we have made some small adjustments to make the design of this figure more in line with the rest of the paper (x-axis separated from the y-axis for example).

Lofverstrom, M., & Liakka, J. (2018). The influence of atmospheric grid resolution in a climate model-forced ice sheet simulation. The Cryosphere, 12(4), 1499-1510.

Lohmann, G., Wagner, A., & Prange, M. (2021). Resolution of the atmospheric model matters for the Northern Hemisphere Mid-Holocene climate. Dynamics of Atmospheres and Oceans, 93, 101206.

Reference

Opsteegh, J. D., Haarsma, R. J., Selten, F. M., and Kattenberg, A.: ECBILT: a dynamic alternative to mixed boundary conditions in ocean models, Tellus A, 50, 348–367, https://doi.org/10.1034/j.1600-0870.1998.t01-1-00007.x, 1998.

---

## Author Comment (AC2)

In the following, the reviewer comment are in blue and our response in black.

Summary:

The study by Quiquet and Roche analyzes various aspects of the climate and ice-sheet evolution in the last two glacial terminations using the intermediate complexity model iLOVECLIM with an interactive ice sheet component. Experiments are presented in which the model is integrated forward from the glacial maximum state (LGM and PGM) through the deglaciation and the interglacial periods. Sensitivity experiments that isolate the influence of individual forcings (e.g., meltwater fluxes, insolation changes, greenhouse gas variations, etc.) are also conducted. The main conclusions are: (i) the Last Interglacial was warmer and had a higher sea-level than the Holocene; (ii) insolation variations is the main driver of glacial retreat during both interglacial periods; (iii) the Atlantic overturning circulation is found to be more sensitive to collapse under Last Interglacial forcing.

The main novelty of the manuscript is the side-by-side comparison of the last two deglaciations in a coupled model setting. However, it is not clear from the presentation what the truly new results are and in what way this study is advancing our understanding of the last two deglaciations. There are several reasons for this, but most importantly because (i) the manuscript does not include a dedicated discussion section where the results are contrasted with the established literature; (ii) the model is quite simplistic and may not be the most appropriate choice for this type of study; (iii) some of the results are undoubtedly model dependent as they contradict previously published results using other models.

I recommend major revisions before this manuscript can be accepted for publication.

Thank you for your thorough review of our manuscript. We have revised the text according to your suggestions. Notably we have added a dedicated discussion section where we discuss simulated atmospheric circulation changes between the LGM and the PGM, using different ice sheet geometries. In this new section, we also discuss our results with respect to the existing literature.

Major comments:

No discussion section:

The lack of a dedicated discussion section makes it hard to get a sense for how the results compare to the established literature and what the potential shortcomings of the study are. You do cite several papers in the results section, but these are primarily used to quantify (and to a certain extent justify) your results. A dedicated discussion section is essential for any study, and this manuscript would certainly benefit from having one as well.

We have added such a section in the revised version of our manuscript.

QGPV model at low resolution:

I wonder how appropriate the model choice is for this study. From reading the model description in Quiquet et al. (2021), the atmospheric component of iLOVECLIM is a spectral, quasi-geostrophic potential-vorticity (QGPV) model that was run at a nominal 5.6-degrees (T21) horizontal resolution.

It seems to me that this model choice is potentially problematic for at least two reasons:

(i) Several studies have shown conclusive evidence that the numerical convergence of both dry and

moist dynamical cores breaks down somewhere between the T31 and T21 resolutions (e.g., Polvani et al. 2004; Lofverstrom and Liakka, 2018), and that resolution can have a substantial influence on the simulated climate (Lohman et al., 2021). The reason for this breakdown is (most likely) that the grid spacing becomes comparable to, or even exceeding the Rossby deformation radius in midlatitudes on sufficiently coarse model grids. This means that baroclinic waves are not appropriately resolved, which are one of the main drivers of the large-scale atmospheric circulation, including the distribution of temperature, precipitation, and wind in mid and high latitudes. While I recognize that it may not be feasible to run the simulations at a different resolution, this potential shortcoming should at least be acknowledged and discussed in the manuscript.

Following your other comment, we have added a discussion section in the revised manuscript, in which we confront our results to existing literature more thoroughly and we present the limitations related to model resolution and physical approximation. We also show the simulated atmospheric circulation differences between the LGM and the PGM.

(ii) I would like to see a thorough discussion on the appropriateness of using a QGPV model as the atmospheric component in a coupled, global model configuration. QGPV is a decent first-order approximation of the synoptic and planetary scale circulation in mid and high latitudes, but it is not an appropriate description of tropical and subtropical circulation where ageostrophic processes dominate because of the smallness of the Coriolis parameter near the equator. Can we really trust a coupled atmosphere-ocean model that is largely incapable of representing the low-latitude atmospheric circulation with even first order accuracy?

The reviewer raises indeed an important limitation of quasi-geostrophic models. However this problem has been identified during the initial development of the model. ECBilt includes ageostrophic terms in the vorticity equation (Opsteegh et al., 1998) that are neglected in the traditional QG approximation. These terms are computed diagnostically from the wind divergence and the tendency of the streamfunction, using an iterative method. We have added this precision of the revised version of the manuscript.

More generally, even with such limitations, this class of models has been proven useful in the past to study global climate dynamics on millenial timescale. An example of such study on the East-Asian Monsoon system (Caley et al., 2014, Nature Comm.) has shown that the model can reproduce some aspects of the multimillenial precipitation evolution in such regions favourably when compared with water isotopologues proxy records. A few additional examples of studies that have used iLOVECLIM or LOVECLIM model and published in highly-cited journals could further include Roche et al. (2004), Renssen et al. (2015), Menviel et al. (2018), Golledge et al. (2019), Menviel et al. (2020), Yin et al. (2021), Park et al. (2023), and many more. Therefore, it is fair to write that such models have been evaluated and confronted to palaeo-data on a range of diverse applications and that their versatility in computing sensitivity experiments renders them somehow more robust than GCMs that have mostly only run time-slices experiments for dedicated time period.

No discussion about atmospheric circulation changes:

Previous studies have shown that the large-scale atmospheric circulation is strongly influenced by both the height and spatial distribution of the Northern Hemisphere ice sheets (e.g., Lofverstrom and Lora, 2017; Kageyama et al. 2021). Importantly, it has been shown that the North American ice sheet affects the temperature and precipitation distribution (i.e., the surface mass balance) over the Eurasian Ice Sheet (e.g, Liakka et al., 2016).

I think this study would be more convincing if the authors also included figures showing changes in

the atmospheric circulation. Not least since the ice-sheet mass balance (i.e., the deglaciation) is to first order driven by changes in the temperature and precipitation distribution, and the QGPV atmospheric model is quite simplistic and may not capture some of the main circulation changes identified in numerous other studies using more comprehensive models.

Atmospheric circulation difference between the LGM and PGM is now shown and discussed with respect to the existing literature.

With respect to the quality of SMB with more comprehensive models, it has been shown on several occasions GCM model outputs are not necessarily appropriate to drive ice sheet models. Using outputs from PMIP3 and PMIP4 model ensemble, both Niu et al. (2019) and van Aalderen et al. (2023) have shown that only a subset of these models were able to maintain reasonable Northern Hemisphere ice sheets at the LGM. The simulated ice sheets at the LGM with iLOVECLIM are not too far from the reconstructions which an indication that the model is not drastically misrepresenting the LGM climate.

Model dependence:

It is compulsory to discuss potential model dependence on results and conclusions in any modeling study. You mention model dependence in a few places in the text, but it would be good to consolidate this in a dedicated discussion section. One of your main conclusions is that insolation is more important for deglaciation than vegetation changes. I agree that this is what your results shows, but it appears to be contradicting the results in, e.g., Sommers et al (2021), who argued that vegetation changes are at least equally important, if not more important than insolation changes for the deglaciation of Greenland in the Last Interglacial. This is just one example of potential model dependence of your results that should be acknowledged and properly discussed in the manuscript.

We agree that we only use one climate model and that our results are representative of this specific model. We do think that inter-model comparison exercises are really useful with this respect so that we can compare model-specific behaviour to more general responses to forcing changes. This is why we have participated to PMIP4 LGM (Kageyama et al., 2021) and deglaciation experiments (in preparation) with our version of the iLOVECLIM model. We follow the same strategy with the GRISLI ice sheet model, participating to the recent ISMIP6, ABUMIP and LarMIP experiments. Outcome of such participations is that our models are not particularly standing out with respect to other participating models.

Unfortunately, there is not yet any intercomparison exercise of coupled ice sheet – climate model simulations of glacial terminations. One reason is that it is far out of reach from most GCM modelling groups at present, but this might change in the future thanks to better computational facilities and improved numerical scaling.

The paper of Sommers et al. (2021) is indeed very relevant since they used a coupled ice sheet – climate model to simulate the last interglacial period. However a direct comparison of this study with our work is not obvious. While we simulate the entire glacial termination, starting from the PGM, Sommers et al. (2021) start their simulation close to the peak insolation of the LIG, at 127 kaBP. Thus they focus on Greenland ice sheet change, not Northern Hemisphere ice sheets. That being said, if our results show that the vegetation change is not the major driver for Northern Hemisphere ice sheet retreat during TII, we nonetheless simulate a larger Greenland ice sheet volume during the LIG when vegetation change is discarded (+20% in ice volume). This is consistent with the work of Sommers et al. (2021). We have added the comparison with this work in the new discussion section of our revised manuscript.

General experiment design:

I am confused by the experiment design. The introduction states that the Northern Hemisphere ice sheet distribution in the PGM and the LGM were quite different, where the former had comparatively more ice in Eurasia relative to the LGM, and vice versa in North America.
However, the experiments presented here use the same ice sheets as initial conditions for both the LGM and the PGM. What is the reason for this choice since this appears to be a substantial deviation from reality? Would a different ice sheet initial condition alter the results in any way, for example through differences in the large-scale atmospheric circulation?
I recognize that spinning up the ice sheets for the PGM is a major task that may be computationally unfeasible. Therefore, I am not necessarily recommending that you re-run the simulations with more appropriate ice sheets for the PGM, but a discussion of the potential influence of these types of deviations from reality should at least be recognized and appropriately discussed in a dedicated discussion section.

Sensitivity experiments with different ice sheets at the LGM (smaller North American ice sheet and larger Eurasian ice sheet) were already part of the initial manuscript. However, it is true that we did not expand too much on the outcomes of these experiments. We have now added more description of the results of these sensitivity experiments. We also discuss the impact of different topographies on the simulated atmospheric circulation.

The motivation for having identical ice sheets as initial conditions for our transient experiments is twofold:
- The extent and size of the PGM ice sheets is a scientific question in itself. There are currently large field-data uncertainties which make these ice sheets a relatively weak target from a modelling point of view.
- Starting from the same ice sheets is a way to more directly quantify the impact of the different external forcings on climate and ice sheets during the last two terminations.

To make it clearer that our model experiments are a simplification of actual past changes we have also added this in the introduction section:
"Using a relatively simplified setup, we do not aim to precisely match the available proxy data but instead we aim at better understanding the role of external forcings (orbital configuration and greenhouse gas concentration) on glacial terminations."

References:

Kageyama et al. (2021). The PMIP4 Last Glacial Maximum experiments: preliminary results and comparison with the PMIP3 simulations. Climate of the Past, 17 (3), 1065–1089. doi: https://doi.org/10.5194/cp-17-1065-2021
Lohmann, G., Wagner, A., & Prange, M. (2021). Resolution of the atmospheric model matters for the Northern Hemisphere Mid-Holocene climate. Dynamics of Atmospheres and Oceans, 93, 101206.
Liakka et al.: The impact of North American glacial topography on the evolution of Eurasian ice sheet over the last glacial cycle, Clim. Past, 1225–1241, https://doi.org/10.5194/cp-12-1225-2016, 2016
Lofverstrom, M., and Liakka, J. (2018). The influence of atmospheric grid resolution in a climate model-forced ice sheet simulation. The Cryosphere, 12(4), 1499-1510
Lofverstrom, M., Lora, J.M., 2017. Abrupt regime shifts in the North Atlantic atmospheric circulation over the last deglaciation. Geophys. Res. Lett. 44, 8047–8055. https://doi.org/10.1002/2017GL074274.

Polvani, L. M., Scott, R., and Thomas, S.: Numerically converged solutions of the global primitive equations for testing the dynamical core of atmospheric GCMs, Mon. Weather Rev., 132, 2539–2552, 2004.
Sommers et al.: Retreat and regrowth of the Greenland ice sheet during the Last Interglacial as simulated by the CESM2-CISM2 coupled climate-ice sheet model. Paleoceanography and Paleoclimatology 36, 2021

**References:**

Caley, T., Roche, D. & Renssen, H. Orbital Asian summer monsoon dynamics revealed using an isotope-enabled global climate model. Nat Commun 5, 5371, https://doi.org/10.1038/ncomms6371, 2014.

Golledge, N.R., Keller, E.D., Gomez, N. et al. Global environmental consequences of twenty-first-century ice-sheet melt. Nature 566, 65–72, https://doi.org/10.1038/s41586-019-0889-9, 2019.

Menviel, L., Spence, P., Yu, J. et al. Southern Hemisphere westerlies as a driver of the early deglacial atmospheric $CO_2$ rise. Nat Commun 9, 2503, https://doi.org/10.1038/s41467-018-04876-4, 2018.

Menviel, L.C., Skinner, L.C., Tarasov, L. et al. An ice–climate oscillatory framework for Dansgaard–Oeschger cycles. Nat Rev Earth Environ 1, 677–693, https://doi.org/10.1038/s43017-020-00106-y, 2020.

Niu, L., Lohmann, G., Hinck, S., Gowan, E. J. and Krebs-kanzow, U. (2019). The sensitivity of Northern Hemisphere ice sheets to atmospheric forcing during the last glacial cycle using PMIP3 models. Journal of Glaciology, 65(252), 645-661. doi:10.1017/jog.2019.42

Opsteegh, J. D., Haarsma, R. J., Selten, F. M., and Kattenberg, A.: ECBILT: a dynamic alternative to mixed boundary conditions in ocean models, Tellus A, 50, 348–367, https://doi.org/10.1034/j.1600-0870.1998.t01-1-00007.x, 1998.

Park, JY., Schloesser, F., Timmermann, A. et al. Future sea-level projections with a coupled atmosphere-ocean-ice-sheet model. Nat Commun 14, 636, https://doi.org/10.1038/s41467-023-36051-9, 2023.

Renssen, H., Mairesse, A., Goosse, H. et al. Multiple causes of the Younger Dryas cold period. Nature Geosci 8, 946–949, https://doi.org/10.1038/ngeo2557, 2015.

Roche, D., Paillard, D. & Cortijo, E. Constraints on the duration and freshwater release of Heinrich event 4 through isotope modelling. Nature 432, 379–382, https://doi.org/10.1038/nature03059, 2004.

van Aalderen, V., Charbit, S., Dumas, C., and Quiquet, A.: Relative importance of the mechanisms triggering the Eurasian ice sheet deglaciation, EGUsphere [preprint], https://doi.org/10.5194/egusphere-2023-34, 2023.

Yin, Q. Z., Wu, Z. P., Berger, A., Goosse, H. & Hodell, D. Insolation triggered abrupt weakening of Atlantic circulation at the end of interglacials. Science 373, 1035–1040, https://doi.org/10.1126/science.abg1737, 2021.

---

## Author Response (AR2)

**Response to Editor Review**

In the following, the editor comments are in blue and our responses in black.

I have gone through your revised version of the manuscript and concluded that it still requires quite some work before publication. I would rate the required revisions as moderate rather than minor. Attached to this decision is the file with your manuscript including comments on the necessary improvements and sample markings of numerous typos and poor formulations using the new Discussion session as an example. I would like to emphasize that the manuscript needs a thorough editing for grammar and clarity, not only in the Discussion section but throughout.

Thank you for your time in reviewing thoroughly our manuscript. We have corrected the typos and formulations you pointed out in the discussion. We have also done our best to improve grammar and clarity for the rest of the paper. We apologize in advance if errors are nonetheless still present and we hope that this does not hamper the understanding. In any case we will be happy to benefit from the Copernicus editing service if our paper gets eventually accepted.

Below are major points that must be improved:
- Model and model setup descriptions (missing details and vague formulations);

Following your suggestion in the annotated pdf we now give more precision on the coupling of the ice sheet model and the the rest of the climate system. Notably, we now provide the equation for surface mass balance and we explain better how the atmospheric temperature bias is indirectly corrected with a geographical adjustment of the melt parameter $c_{rad}$ in the melt equation.

We also corrected the imbalance between the description of the climate model and the ice sheet model, simplifying the description of the ice sheet model since we do not do any modification to it for this paper.

We have added a table in the supplement that contains the major parameters of the coupled ice sheet – climate model. We have also added a Supporting Text to explain more precisely how the alternative ice sheets were elaborated (stand-alone climate and ice sheet integrations). This reads:

"In the standard version of the model, the melt parameter $c_{rad}$ used in the surface mass balance model is locally changed according to the annual mean temperature bias with respect to present-day reanalysis ERA-interim. The local modification is linear, using a coefficient of $0.1°C^{-1}$. This has been implemented to indirectly correct for the temperature biases in the climate model. For a temperature bias of +10°C we use a $c_{rad}$ of -80 W m$^{-2}$ instead of the reference value of -40 W m$^{-2}$.

To elaborate alternative ice sheet geometries for the penultimate glacial maximum, we run 100-yr long simulations of the climate model with prescribed LGM ice sheets branched from the 142 kaBP climate equilibrium. For theses simulations we modify regionally the value of the reference temperature bias in order to impact the value of the local melt parameter value. For the two alternative ice sheet geometries, we divide by 5 the temperature bias in the region of North America (approximatively for longitudes from -140 to 0°E). Since the temperature bias is mostly positive in this region, this modification results in higher value of the $c_{rad}$ parameter (more melt). For Eurasia (longitude lower to about 140°E), we replace the temperature bias by a value of +20°C (larger Eurasian ice sheet case) and +40°C (much larger Eurasian ice sheet case). These modifications in Eurasia produce a larger SMB. Finally, we use the climatological SMB resulting from these 100-yr

long simulations to force offline the ice sheet model until equilibrium, similarly to what we did to generate the initial LGM ice sheets."

- Justification of the initial ice sheet geometry choices;

It is true that in the original version of the paper we only referred to our previous publication on the last deglaciation. To make the manuscript self-consistent we now explain how we elaborated our initial reference ice sheet geometry for TI, which is the same as for TII, in the beginning of the experimental setup section (Sec. 2):

"The experiments discussed here for TI are the coupled ice sheet – climate model simulations covering the last 26 kaBP from Quiquet et al. (2021). For these, the initial climate conditions and ice sheet geometries were obtained using uncoupled simulations. We first run the climate model for 3,000 years with prescribed ice sheet reconstructions (GLAC-1D, Tarasov et al., 2012; Tarasov and Peltier, 2002; Briggs et al., 2014) using fixed 21 kaBP orbital configuration (Berger, 1978) and greenhouse gas forcings (Lüthi et al., 2008). The last hundred years of this climate spin-up is used to derive climatological climate forcings required by the ice sheet model. We used these forcings to run stand-alone ice sheet model simulations for 200-kyr to reach equilibrium. The spun-up ice sheet and climate states were then used as initial conditions for our coupled simulations."

Later when we describe the different experiments performed we now provide technical details of how we elaborate the two alternative initial ice sheets that we used for TII. Since there is no evidence for a significantly different eustatic sea level for the PGM with respect to the LGM, our alternative ice sheets do not imply large changes in total ice volume. The first alternative (slightly smaller NAIS, -6%, and larger EIS, +36%) does not change the total ice volume stored on land. The second alternative (slightly smaller NAIS, -6%, and much larger EIS, +71%) corresponds to an increase of about 5~m of sea level equivalent of the total ice volume. This part now reads:

"Accelerated experiments are first used to assess the sensitivity of the simulated TII to the initial ice sheet geometry. Our initial ice sheet geometry for our TII experiment is the same as for the TI experiment. This is a modelling simplification since it is unlikely that the configuration of the Northern Hemisphere ice sheets was identical for the two previous glacial maximums. To explore this model assumption, we elaborated alternative PGM ice sheet geometries. To generate these we run new stand-alone ice sheet model simulations using different SMB forcings to the ones used to generate the LGM ice sheet spin-up. The new SMB forcings were obtained by running the climate model for 100-yr simulations with regionally modified crad parameter in the melt equation of the ITM. In the reference model, this parameter is locally adjusted to indirectly correct for the temperature bias in the model. To obtain alternative SMB we apply regional modifications to this temperature bias. More specifically, we reduce the bias correction in North America in order to generate higher surface melt rates since the temperature bias is positive in this region. In Eurasia, we impose a fixed artificial positive bias so that the crad gets reduced to produce less melt. More information on these modifications is available in the supplement (Supp. Text 1). These artificial SMB modifications are only used to produce alternative ice sheets with GRISLI stand-alone simulations but they are removed for transient coupled simulations. The alternative ice sheet geometries consist in a reduced North American ice sheet by about 6% in volume with respect to the LGM (about -2.0 $10^6$ km$^3$) and a larger (+36% volume, about +2.1 $10^6$ km$^3$) or much larger (+71%, about +4.2 $10^6$ km$^3$) Eurasian ice sheet. The first alternative (larger Eurasian ice sheet) does not change the total ice volume stored on land while the second (much larger Eurasian ice sheet) corresponds to an increase of about 5 m of sea level equivalent of this volume. The alternative Eurasian ice sheets display a larger extent towards the East more in agreement with the palaeo data

(Svendsen et al., 2004). These experiments serve to quantify the sensitivity of our simulated deglacial climate and ice sheet trajectories to the ice sheet glacial geometry."

One point we need to mention is that it is difficult to use very different ice sheets while keeping some realism for the deglaciation. We made a lot of different tests varying ice sheet geometries but due to the strong albedo feedback in coupled simulations it is very easy to end up with very different climate trajectories for all these tests. For example, larger Eurasian ice sheets tend to easily expand to the East, generating a large Siberian ice sheet which is not present in our reference simulations. In North America it happens very often that the ice sheet expands too much over Alaska, making eventually a bridge between North America and Siberia. This feature is also found in other experiments by other groups (Willeit and Ganopolski, 2018), even with more complex models (e.g. Ziemen et al., 2014), but is known to be inconsistent with ice-sheet reconstructions. These ice sheets can become very resilient and they can survive a deglaciation due to the strong albedo feedback. We decided not to keep all these different geometries since they are too far away from our general understanding of ice sheet geometry changes through the two last glacial-interglacial cycles. Instead we have preferred to use relatively small changes with respect to the LGM configuration.

- A stronger discussion of model limitations and their implications (one of the reviewer's comments has not been addressed sufficiently – regarding QGPV model at low resolution, review 2);

Reviewer 2 is correct in suggesting that QGPV approximations in atmospheric models is intrinsically inadequate to simulate parts of the tropical atmospheric circulation. This is a well-acknowledged limitation of QGPV model and that is why, in the early development of ECBilt (the atmospheric component of iLOVECLIM), additional ageostrophic terms have been added as potential vorticity forcings (cf. Opsteegh et al., 1998). This particularity might explain why the model has been shown able to reproduce to a first order some aspects of the tropical climate (Goosse et al., 2010), such as the East-Asian monsoon activity, even in its broad late Quaternary evolution when compared to water isotopologues proxy record (Caley et al., 2014).

This information was already added in our revised version of the manuscript. To be completely honest we do not know exactly what we should add with this respect.

If taken to face value, the comment of Reviewer 2 implies that any model that is not implementing the primitive equations to its fullest should not be used in coupled climate studies. This is both excessive and very limiting. Excessive in the sense that, at present, there are still many other models that implement various forms of the equations of the atmospheric circulation that are not the full implementation of the primitive equations. One can, for example, think about the statistical-dynamical atmospheric models class (cf. Petoukhov, 2003, POTDSAM ; Totz et al., 2018, AEOLUS1.0) or models that do implement the primitive equations in other forms (AEOLUS2.0, https://www.pik-potsdam.de/en/institute/departments/earth-system-analysis/models/aeolus-2.0) that are argued to be more pertinent for some aspects of the atmospheric variability (Rostami et al., 2022). Very limiting in the sense that even the appropriate discretization of the primitive equations can be disputed: should there be a limit on the number of vertical layers to describe the troposphere – stratosphere interaction? Are models such as simplified GCMs (Molteni, 2003, Kucharski et al., 2006, SPEEDY model ; Smith et al., 2008 & Smith, 2012, FAMOUS model) also too limited for this? Even further, one could argue that GCMs are also very inadequate to represent many aspects of the weather at fine scales, contributing to the climate potentially and that CRMs should be used (Stevens et al., 2020), while some others still argue to keep the GCM class (Balaji et al., 2022).

The latter type of reasoning is always pushing to have "more" and results in models that are way to computationally demanding to be useful at the timescales we are looking at in our manuscript, such that intermediate complexity retain its value (Kucharski et al., 2013).

Overall, the recent past record of publications in the field has shown that many aspects of the climate system, on a long-term basis, can be investigated with such models as the ones we cite here-above. Though not representing all aspects of the climate as one could wish for, they are still a valued tool to access realms that are not even accessible with GCMs.

The remark of the reviewer is an extremely general question that casts doubt on many results obtained with such kind of models over more than 40 years of scientific research. There is no study, to our knowledge, that show that the approximation made in our atmospheric model and others should dismiss the general conclusions we reach in our study. We thus do not feel that our paper should contain a dedicated paragraph on this matter since it is far beyond the target of our manuscript.

- Discussion of some earlier studies is too superficial;

We have largely rewritten the discussion section, in particular the limitations due to atmospheric model resolution. This section now reads:

"The simulated atmospheric circulation changes when using different ice sheet geometries at the PGM do not seem to impact drastically the individual ice sheet volume evolution through TII (Fig. S7). These can be caused by the low spatial resolution of our atmospheric model that can underestimate the atmospheric circulation changes. For example, Lofverstrom and Liakka (2018) used an atmospheric-only general circulation model at various spatial resolutions to generate climate forcings to run stand-alone ice sheet model simulations. They showed that the model ability to reconstruct the LGM ice sheets strongly depends on the spatial resolution of the atmospheric model, higher resolution showing generally better performance. The authors suggest in particular that the T21 spatial resolution is fundamentally inadequate to resolve numerically the baroclinic waves. Indeed, to insure stability of the numerical scheme, coarse resolution models show a larger diffusivity which dampens the waves (Magnusdottir and Haynes, 1999; Polvani et al., 2004; Lofverstrom and Liakka, 2018). However, while we use a T21 resolution, our model temperature biases are not comparable to the ones shown in Lofverstrom and Liakka (2018). For example, they show that their model at T21 is unable to reconstruct the Eurasian ice sheet, independently from the surface mass balance scheme they use. In our case, the model does build up an ice sheet in Western Eurasia and none in Siberia, even without the indirect bias correction that we use in the melt equation (Eq. 2 leads to increase crad in Eurasia, inducing more melt). This suggests that other biases (apart from numerical diffusion) can alter model performance and that the fact that our model correctly represents the LGM ice sheets might be the results of some compensating biases. More generally, using outputs from the Paleoclimate Modelling Intercomparison Project (PMIP) phase 3 and 4 LGM database to force ice sheet models, both Niu et al. (2019) and van Aalderen et al. (2023) show that most general circulation models do not provide suitable climatic forcing fields to reconstruct ice sheets in agreement with geological reconstructions. These deficiencies are generally not related to spatial resolution differences amongst participating models. However, for a given climate model, a higher spatial resolution will tend to have a more accurate representation of the topography and this will induce noticeable difference with its lower spatial resolution version (Lohmann et al., 2021). In fact, SMB is highly correlated to topography, notably due to the direct impact of elevation on surface temperature. This is why different groups have used different strategies to downscale ice-processes (Robinson et al., 2010; Fyke et al., 2011; Krebs-Kanzow et al. 2021; Crow et al., 2024). While the downscaling scheme that we use does not allow any improvement in the topographically-induced atmospheric circulation change, it nonetheless better capture the melt elevation feedback than a standard vertical lapse rate approach"

- The literature overview can be improved through citations of more studies presenting geological evidence of deglaciation modes through TI vs TII.

There are no strong evidence for the deglaciation pattern of the Northern Hemisphere ice sheets during TII (Pollard et al., 2023). This lack of constraints was already noted in the community modelling protocol of Menviel et al. (2019). The reconstructions of ice sheet during the PGM are also subject to considerable uncertainties, notably since the maximal ice sheet extension is not necessarily synchronous between the different ice sheets. Some regions of the Eurasian ice sheet show a maximal extent circa 160 kaBP, much earlier than the PGM (Hughes and Gibbard, 2018; Pollard et al., 2023).

We have added this precision in the introduction when we discuss the ice sheet geometry differences between the PGM and LGM:
"Nevertheless, the maximal expansion of the Eurasian ice sheet might have occurred significantly earlier than the PGM (Hughes and Gibbard, 2018; Pollard et al., 2023) and precise reconstruction of the PGM ice sheets is still lacking."

More generally we have added a few additional reference for proxy data in the introduction. At the end of the paragraph discussing TII with respect to the palaeo records:

"Other types of records, such as speleothems or oceanic sediment data, display abrupt changes, concomitant with oceanic changes (Martrat et al., 2014; Govin et al., 2015; Cheng et al., 2016)."

And later, we added a new paragraph dedicated to a more direct inter-comparison of the two terminations. This paragraph reads:

"In summary, the last two glacial terminations display significant differences. In terms of ice sheet disintegration, there are some proxy data evidence for a higher rate of mass loss during TII with respect to TI (Carlson, 2008; Stoll et al., 2022; Grant et al., 2014). This higher loss rate might explain the long (~7 ka) period of weak AMOC across TII (Böhm et al., 2015; Deaney et al., 2017). A feature that significantly differs from the several shorter events during TI (McManus et al., 2004). If speleothem and oceanic records suggest that H11 share similar large scale characteristics with H1 or the Younger Dryas, these events largely differ in terms of timing of their occurrence during the termination (Martrat et al., 2014; Govin et al., 2015). In terms of ice sheet geometries, apart from the fact that they were different for the two glacial maximums (Svendsen et al., 2004; Pollard et al., 2023), the geometry changes through the terminations cannot be easily compared due to the lack of strong constraints for TII."

With this decision, I encourage you to rework your manuscript and submit it for final decision.

**References**

Balaji, V., Couvreux, F., Deshayes, J., Gautrais, J., Hourdin, F., & Rio, C. (2022). Are general circulation models obsolete? *Proceedings of the National Academy of Sciences*, 119(47), e2202075119. https://doi.org/10.1073/pnas.2202075119

Caley, T., Roche, D. M., and Renssen, H.: Orbital Asian summer monsoon dynamics revealed using an isotope-enabled global climate model, Nature Communications, 5, 1–6, https://doi.org/10.1038/ncomms6371, 2014.

Goosse, H., Brovkin, V., Fichefet, T., Haarsma, R., Huybrechts, P., Jongma, J., Mouchet, A., Selten, F., Barriat, P.-Y., Campin, J.-M., Deleersnijder, E., Driesschaert, E., Goelzer, H., Janssens, I., Loutre, M.-F., Morales Maqueda, M. A., Opsteegh, T., Mathieu, P.-P., Munhoven, G., Pettersson, E. J., Renssen, H., Roche, D. M., Schaeffer, M., Tartinville, B., Timmermann, A., and Weber, S. L.: Description of the Earth system model of intermediate complexity LOVECLIM version 1.2, Geosci. Model Dev., 3, 603–633, https://doi.org/10.5194/gmd-3-603-2010, 2010.

Kucharski F, Molteni F, and Bracco A (2006) Decadal interactions between the western tropical Pacific and the North Atlantic Oscillation. Clim Dyn 26: 79-91

Kucharski, F., F. Molteni, M. P. King, R. Farneti, I. Kang, and L. Feudale, 2013: On the Need of Intermediate Complexity General Circulation Models: A "SPEEDY" Example. *Bull. Amer. Meteor. Soc.*, **94**, 25–30, https://doi.org/10.1175/BAMS-D-11-00238.1.

Molteni F (2003) Atmospheric simulations using a GCM with simplified physical parametrizations. I. Model climatology and variability in multi-decadal experiments. Clim Dyn 20: 175-191

Opsteegh, J. D., Haarsma, R. J., Selten, F. M., and Kattenberg, A.: ECBILT: a dynamic alternative to mixed boundary conditions in ocean models, Tellus A, 50, 348–367, https://doi.org/10.1034/j.1600-0870.1998.t01-1-00007.x, 1998.

Petoukhov V., Ganopolski A. and Claussen M., 2003. POTSDAM - a set of atmosphere statistical-dynamical models: Theoretical background. PIK Report No. 81, Potsdam Institute for Climate Impact Research, Potsdam, Germany, 136 pp.

Pollard, O. G., Barlow, N. L. M., Gregoire, L. J., Gomez, N., Cartelle, V., Ely, J. C., and Astfalck, L. C.: Quantifying the uncertainty in the Eurasian ice-sheet geometry at the Penultimate Glacial Maximum (Marine Isotope Stage 6), The Cryosphere, 17, 4751–4777, https://doi.org/10.5194/tc-17-4751-2023, 2023.

Rostami, M., Zhao, B. & Petri, S.: On the genesis and dynamics of Madden–Julian oscillation-like structure formed by equatorial adjustment of localized heating, Quarterly Journal of the Royal Meteorological Society, 148(749), 3788–3813, 2022.

Smith, R. S., Gregory, J. M., and Osprey, A.: A description of the FAMOUS (version XDBUA) climate model and control run, Geosci. Model Dev., 1, 53–68, https://doi.org/10.5194/gmd-1-53-2008, 2008.

Smith, R. S.: The FAMOUS climate model (versions XFXWB and XFHCC): description update to version XDBUA, Geosci. Model Dev., 5, 269–276, https://doi.org/10.5194/gmd-5-269-2012, 2012.

Stevens, B., Acquistapace, C., Hansen, A., Heinze, R., Klinger, C., Klocke, D., Rybka, H., Schubotz, W., Windmiller, J., Adamidis, P., Arka, I., Barlakas, V., Biercamp, J., Brueck, M., Brune, S., Buehler, S. A., Burkhardt, U., Cioni, G., Costa-Suros, M., Crewell, S., Crüger, T., Deneke, H., Friederichs, P., Henken, C. C., Hohenegger, C., Jacob, M., Jakub, F., Kalthoff, N., Köhler, M., van Laar, T. W., Li, P., Löhnert, U., Macke, A., Madenach, N., Mayer, B., Nam, C., Naumann, A. K., Peters, K., Poll, S., Quaas, J., Röber, N., Rochetin, N., Scheck, L., Schemann, V., Schnitt, S., Seifert, A., Senf, F., Shapkalijevski, M., Simmer, C., Singh, S., Sourdeval, O., Spickermann, D., Strandgren, J., Tessiot, O., Vercauteren, N., Vial, J. Voigt, A., and Zängl, G.: The added value of large-eddy and storm-resolving models for simulating clouds and precipitation, J. Meteorol. Soc. Jpn., 98, 395–435, 2020a.

Totz, S., Eliseev, A. V., Petri, S., Flechsig, M., Caesar, L., Petoukhov, V., and Coumou, D.: The dynamical core of the Aeolus 1.0 statistical–dynamical atmosphere model: validation and parameter optimization, Geosci. Model Dev., 11, 665–679, https://doi.org/10.5194/gmd-11-665-2018, 2018.

Willeit, M. and Ganopolski, A.: The importance of snow albedo for ice sheet evolution over the last glacial cycle, Clim. Past, 14, 697–707, https://doi.org/10.5194/cp-14-697-2018, 2018.

Ziemen, F. A., Rodehacke, C. B., and Mikolajewicz, U.: Coupled ice sheet–climate modeling under glacial and pre-industrial boundary conditions, Clim. Past, 10, 1817–1836, https://doi.org/10.5194/cp-10-1817-2014, 2014.